# CABS: CONFLICT-AWARE AND BALANCED SPARSIFICATION FOR ENHANCING MODEL MERGING

## ABSTRACT

Model merging based on task vectors, i.e., the parameter differences between fine-tuned models and a shared base model, provides an efficient way to integrate multiple models without retraining. This approach can be used to combine task-specific models into a multitask model, improve generalization, or address model deficiencies. One of the significant challenges faced by model merging is the conflicts between task vectors. Existing works aim to mitigate these conflicts through sparsification; however, two issues observed in our experiments significantly limit their performance: *high parameter overlap* and *unbalanced weight distribution*. To address these issues, we propose a simple yet effective framework called CABS (Conflict-Aware and Balanced Sparsification), consisting of **C**onflict-**A**ware Sparsification (CA) and **B**alanced **S**parsification (BS). CA can reduce parameter overlap by applying masks during sequential pruning, ensuring that each task vector retains distinct, non-overlapping parameters. BS leverages $n$:$m$ pruning to preserve critical weights while maintaining an even distribution across layers. Our comprehensive experiments demonstrate that CABS outperforms state-of-the-art methods across a range of diverse tasks and model sizes. Notably, in experiments with 7B-parameter language models, CABS surpasses the average performance of an "ideal" model, a virtual model that selects the highest score from individual fine-tuned models for each task (CABS: 76.50 vs. Ideal Model: 76.30 vs. Baseline: 76.02 vs. Fine-tuned Model: 75.86). Our results highlight the importance of addressing both high parameter overlap and unbalanced weight distribution to achieve robust and high-performance model merging.

## 1 INTRODUCTION

Model merging has gained increasing attention in the deep learning community, particularly in the context of using task vectors for model merging in large language models (LLMs) (Ilharco et al., 2022; Li et al., 2023; Wortsman et al., 2022; Jin et al., 2022; Matena & Raffel, 2022; Singh & Jaggi, 2020; Akiba et al., 2024). This technique has become especially popular for merging homologous models—those fine-tuned from the same base models—to create better-performing models. Many top-performing models on the LLM leaderboard (Beeching et al., 2023) are built by fine-tuning base models and subsequently merging them to optimize task-specific performance. Additionally, major enterprises have employed model merging techniques in the development of pretraining models, such as LLaMA3 (Dubey et al., 2024) and Qwen2 (Yang et al., 2024; Lu et al., 2024), to enhance generalization capabilities and improve performance across a range of tasks.

Recent studies have further shown that sparsifying task vectors before merging can mitigate parameter conflicts between different task vectors, leading to measurable improvements in merging performance (Yu et al., 2024; Yadav et al., 2024; Davari & Belilovsky, 2023; He et al., 2024). These conflicts can be categorized into two types: (a) conflicts due to redundant parameters, where parameters that contribute little to performance are unnecessarily retained, and (b) conflicts due to overlapping parameters, where task vectors retain parameters that overlap, potentially with significantly different magnitudes or signs. These overlaps make the merging process less efficient.

Sparsifying can be achieved by selectively or randomly dropping part of a task vector. This process is similar to one-shot pruning, with the former aiming to reduce conflicts in model merging and the latter targeting model compression. Magnitude-based pruning (Liang et al., 2021) is one of the

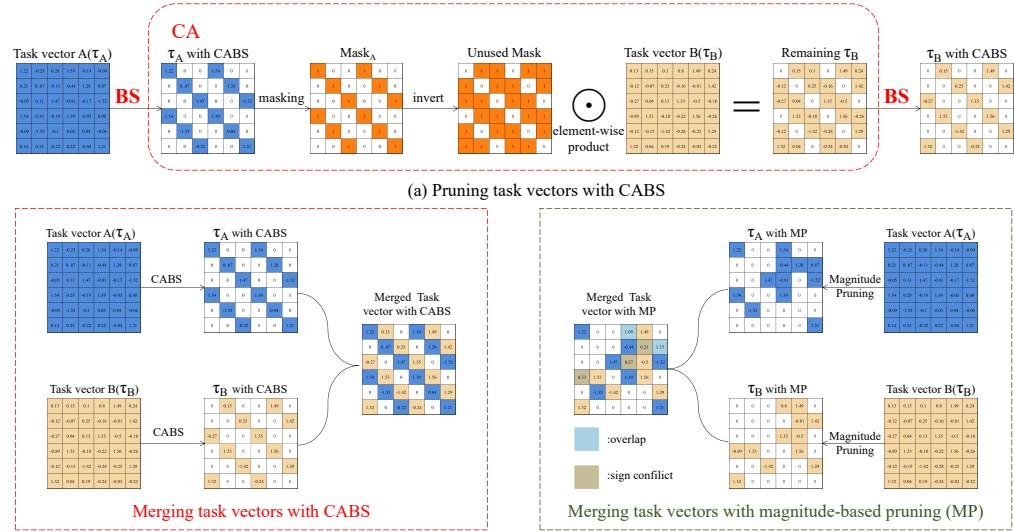

(a) Pruning task vectors with CABS

(b) Merging task vectors with and without CABS

Figure 1: Illustration of the **CABS** framework, which enhances model merging by addressing parameter overlap and weight imbalance. By integrating Conflict-Aware Sparsification (CA) and Balanced Sparsification (BS), CABS delivers more effective merging compared to standard merging with magnitude-based pruning (MP), leading to improved model performance.

mainstream pruning techniques, which can efficiently estimate the importance of weights and selectively preserve the essential weights, thus rightfully superior to random pruning. Inspired by pruning techniques, recent model merging studies (Yadav et al., 2024) applied magnitude-based pruning for sparsifying task vectors with the important weights retained. However, as pointed out by DARE (Yu et al., 2024), the results are counterintuitive —magnitude-based pruning underperforms compared to random weight-dropping methods. This unexpected phenomenon contradicts the observations of widely studied pruning techniques, which demonstrate that retaining important weights helps preserve model performance.

Our research explores the reasons behind this discrepancy, especially in a setting where magnitude-based pruning is expected to perform well. Addressing these issues is key to advancing model merging and developing high-performance merged models. Specifically, by analyzing the weight distribution and overlap in task vectors produced by DARE and magnitude-based pruning, we identified two key factors contributing to the underperformance of magnitude-based pruning:

**High Parameter Overlap**: After magnitude-based pruning, the retained weights of different task vectors often exhibit significant overlap, particularly compared to random methods like DARE. This leads to increased conflicts between task vectors during model merging, ultimately affecting the resulting model performance.

**Unbalanced Weight Distribution**: Magnitude-based pruning tends to distribute retained weights unevenly across the model's weight matrices, with some regions retaining significantly more weights than others. After pruning, the model merging process applies a uniform rescaling factor globally across the model to restore performance. However, this process amplifies the existing imbalance, ultimately leading to suboptimal performance. In contrast, random pruning methods like DARE can avoid this problem, which maintain better balance across the model by distributing weights more uniformly.

To address the issues uncovered above, we propose a novel framework: **Conflict-Aware and Balanced Sparsification (CABS)**. As illustrated in Figure 1, CABS distinguishes itself from existing methods by introducing two key strategies:

**Conflict-Aware (CA) Sparsification**: CA addresses conflicts between task vectors by employing a sequential pruning approach, ensuring *no overlap* between the retained weights of different task vectors. As shown in Figure 1 (a), CA first applies pruning to task vector A (blue, $\tau_A$), and then masks the overlapping weights when pruning task vector B (yellow, $\tau_B$). This masking technique

minimizes conflicts during the merging process by removing shared weights, allowing for more effective task vector merging and improving the final model performance.

**Balanced Sparsification (BS)**: BS addresses the issue of unbalanced weight distribution by applying n:m pruning, which selectively retains $n$ weights out of every $m$ consecutive weights based on magnitude (Zhou et al., 2021). As demonstrated in Figure 1 (a), BS is applied first to $\tau_A$, followed by another application to Remaining $\tau_B$ after CA has eliminated overlapping weights. This ensures a more uniform distribution of weights across layers, reducing the adverse effects of weight concentration in certain regions.

These strategies are straightforward, yet highly effective. Our extensive experiments on both decoder-based Mistral-7B (Jiang et al., 2023) and encoder-based RoBERTa-Base (Liu, 2019) models, spanning tasks from the LLM leaderboard and the GLUE (Wang et al., 2018) dataset respectively, demonstrate that CABS effectively addresses the issues associated with magnitude-based pruning. In Mistral-7B experiments, CABS achieved an average performance of 76.50, outperforming the "ideal" virtual model (76.30), which is a hypothetical model that picks the highest score from each fine-tuned model for every task. with previous SOTA methods scoring 76.02 and fine-tuned models at 75.86. In RoBERTa-Base experiments, CABS improved task performance to 81.49, outperforming previous SOTA method (80.65) and the baseline task-arithmetic score (80.15). While absolute improvements may appear small, they consistently confirm CABS's superiority across different architectures. Furthermore, an ablation study verifies the validity of each strategy.

**Our contributions are as follows:**

- We identify two key issues encountered by magnitude-based pruning in the context of task vector sparsification, i.e., high parameter overlap and unbalanced weight distribution.

- We propose the CABS framework, consisting of conflict-aware sparsification and balanced sparsification strategies, which can effectively address the two identified issues.

- We conduct comprehensive experiments across a variety of tasks and model sizes, showing that CABS outperforms state-of-the-art methods.

- We are the first to introduce an "ideal" yet rigorous baseline for evaluation, where CABS outperforms this virtual baseline while all existing methods fall short.

Resources and implementation details of our approach are available at `https://anonymous.4open.science/r/CABS-027B`.

## 2 RELATED WORK

**Model merging** has become a vital strategy for combining multiple fine-tuned models into a single multitask model without requiring additional training. Fine-tuned models from the same pre-trained model often share part of the optimization trajectory, making them suitable for merging. This process can enhance performance on target tasks, improve out-of-domain generalization, and support applications such as federated learning, model compression, and continual learning.

The simplest merging technique involves directly averaging the model parameters (Izmailov et al., 2018; Wortsman et al., 2022). However, this naive approach often fails to account for task-specific variations, leading to suboptimal performance. A more refined approach, **Task Arithmetic** (Ilharco et al., 2022), was introduced as a pioneering method in the realm of task vector-based merging. In Task Arithmetic, task vectors—computed as the difference between fine-tuned model parameters and their initial pre-trained values—are combined using weighted sums to create a multitask model. However, it struggle with issues such as parameter redundancy and sign conflicts.

To address some of these issues, **TIES-Merging** (Yadav et al., 2024) introduces a more sophisticated approach that operates in two key ways: first, by pruning parameters that are not significantly impactful, thereby reducing the influence of redundant parameters; and second, by resolving sign conflicts during the merging process. This dual approach minimizes interference between task vectors and ensures that the most critical parameters are preserved and properly aligned during the merge. **DARE** (Yu et al., 2024), a technique inspired by **Dropout** (Srivastava et al., 2014), reveals the high redundancy in task vectors by randomly dropping 90% of the parameters and rescaling the remaining

ones. Using random pruning, DARE has been shown to outperform magnitude-based pruning methods in model merging. However, DARE does not fully explain the reasons for this improvement. Our analysis suggests that DARE helps mitigate some of the overlap and imbalance. Nevertheless, the random nature of the approach may potentially sacrifice precision. Other categories approaches, such as **Evolutionary Model Merge**(Akiba et al., 2024) and **Pack of LLMs**(Mavromatis et al., 2024), are detailed in the AppendixA.17.

**Network pruning techniques,** particularly **magnitude pruning** (Zhu & Gupta, 2018), have been extensively studied for their role in optimizing model performance and reducing computational costs (Liu et al., 2019; Frankle & Carbin, 2018; Gale et al., 2019; Zhu & Gupta, 2018). Magnitude pruning retains parameters based on their magnitude, assuming that larger magnitudes correspond to more critical information (Kovaleva et al., 2021; Puccetti et al., 2022; Yin et al., 2023). However, when applied in the context of model merging, this approach can lead to an unbalanced distribution of retained weights, which exacerbates conflicts during the merging process and results in suboptimal performance.

To address this issue, while **n:m pruning** (Zhou et al., 2021; Xia et al., 2022)was originally designed for structured pruning and inference acceleration, we discovered that it can be repurposed to control the balance of sparsified task vectors in model merging. Although n:m pruning may not perform as well as unstructured pruning in traditional scenarios, our findings demonstrate that it effectively mitigates weight imbalance, leading to improved performance in merged models. This insight forms a key contribution of our work, highlighting the potential of n:m pruning in enhancing model merging outcomes.

Our proposed **CABS** method builds upon prior works by introducing CA, a novel approach designed to eliminate parameter overlap during model merging. Additionally, it repurposes the existing n:m pruning technique to mitigate unbalanced weight distribution. Together, CABS effectively enhances the stability and performance of model merging.

## 3 ISSUES IN TASK VECTOR SPARSIFICATION FOR MODEL MERGING

In model merging, particularly when using sparse task vectors to combine models fine-tuned for different tasks, an unexpected phenomenon has emerged: magnitude-based pruning, which typically retains weights with larger absolute values, often underperforms compared to random pruning methods like DARE (Yu et al., 2024). This result contradicts the intuition that preserving critical knowledge, rather than randomly retaining information, within the task vectors should enhance the performance of the merged model. Our investigation into this phenomenon reveals two key issues: the overlap between retained weights and their unbalanced distribution within each task vector.

**High Parameter Overlap.** By comparing the overlap rate between magnitude-based and random pruning methods, our analysis demonstrates that magnitude-based pruning results in a significantly higher parameter overlap between task vectors compared to random pruning methods. As shown in Figure 2, although the overlap rate of magnitude-pruned task vectors decreases gradually with increasing sparsity, it remains significantly higher than that of randomly pruned vectors, especially at higher sparsity levels. This disparity highlights the key issue with magnitude-based pruning, where high overlap persists even as the model becomes sparser.

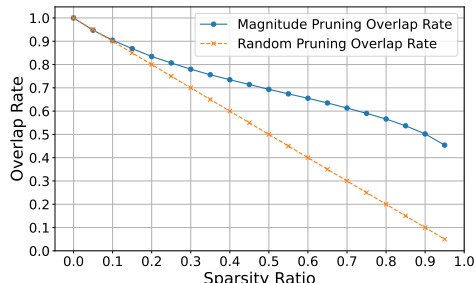

Figure 2: The trend of overlap rate along the sparsity ratio shows that the overlap rate achieved by magnitude-based pruning decreases more slowly than that of random pruning, with the gap widening progressively.

This elevated overlap in magnitude-pruned vectors introduces conflicts during model merging, as overlapping parameters may have significantly different magnitudes or signs between task vectors. These conflicts reduce the efficiency of the merging process and hinder the model's ability to perform optimally on individual tasks,

ultimately leading to suboptimal task-specific performance. The performance implications of these overlapping parameters are explored in detail in 5.4. For details on how the overlap rate is calculated, please refer to Appendix A.1.

**Unbalanced Weight Distribution.** By visualizing the weight distribution shown in Figure 3, we identified another critical issue: the unbalanced distribution of retained weights caused by magnitude-based pruning. Magnitude pruning often leads to weight concentration in specific regions of the model's weights. This imbalance is further exacerbated by the rescaling process, where certain weights gain disproportionate influence over the model's output, often resulting in suboptimal performance. This uneven distribution is particularly detrimental after sparsification, as it hampers the merged model's ability to generalize effectively. The

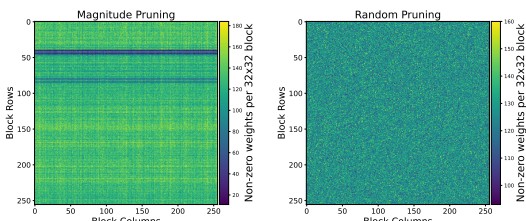

Figure 3: Magnitude pruning results in a more concentrated and unbalanced distribution of weights compare to random pruning.

performance implications of these unbalanced weights are discussed in detail in 5.4.

To comprehensively analyze this issue, we further examined the weight distributions across different layers of the model, including the query-key-value (QKV) projection and MLP layers, at various sparsity levels (e.g., 50%, 75%, and 90%). These experimental results are provided in Appendix A.2, demonstrating the pervasive nature of the imbalance across different layers and sparsity levels.

## 4 METHODOLOGY

### 4.1 OVERVIEW OF CABS FRAMEWORK

To address the aforementioned issues, we propose the CABS (Conflict-Aware and Balanced Sparsification) framework. As illustrated in Figure 1, CABS resolves parameter conflicts and ensures balanced weight distribution so as to enhance the performance of the merged model. The framework integrates two core strategies: Conflict-Aware Sparsification (CA) and Balanced Sparsification (BS), which will be detailed in the following sections. Algorithm 1 demonstrates how these strategies are implemented in CABS.

---
**Algorithm 1** CABS
---
**Input:** Task vectors $\tau_A, \tau_B$, base model $W_{\text{base}}$, sparsity level $n$ , $m$, rescale factors $\lambda_A$ , $\lambda_B$
**Output:** Parameters of the merged model $W_{\text{final}}$
  1: Apply n:m pruning to $\tau_A$ and compute $\text{mask}_A$                 // include BS
  2: $\tau_{\text{B remaining}} = \tau_B \odot (1 - \text{mask}_A)$ to eliminate overlap with $\tau_A$     // core step of CA
  3: Apply n:m pruning to $\tau_{\text{B remaining}}$ to compute $\text{mask}_B$          // include BS
  4: Merge the pruned vectors with the base model:
$$W_{\text{final}} = W_{\text{base}} + \lambda_A \times \text{mask}_A \odot \tau_A + \lambda_B \times \text{mask}_B \odot \tau_B$$
  5: **return** $W_{\text{final}}$

---

### 4.2 CONFLICT-AWARE SPARSIFICATION (CA)

**Motivation**. During model merging, overlapping task vectors can lead to performance degradation of the merged model because different task vectors may update the same parameters inconsistently, often with differing magnitudes or signs.By minimizing these overlaps, it is expected to enhance the stability and performance of the merged model.

**Sequential Pruning and Mask Application**. CA aims to eliminate parameter overlap during model merging by employing a sequential pruning strategy. The process begins with the first vector $\tau_A$ being pruned, producing a mask $mask_A$ that marks the positions of the retained weights. This mask is then used to guide the pruning of the second task vector $\tau_B$, ensuring that there is no overlap between the parameters of $\tau_A$ and $\tau_B$.

For the second task vector $\tau_B$, the prior mask $mask_A$ is applied in an inverted form to determine the remaining weights that do not overlap with the first pruned task vector. Specifically, the remaining weights of $\tau_B$ are calculated as:

$$\tau_{\text{B remaining}} = \tau_B \odot (1 - \text{mask}_A).$$

This ensures that only the non-overlapping weights in $\tau_B$ are retained in the subsequent pruning process. Afterward, a second round of pruning is performed on $\tau_{\text{B remaining}}$, generating a new sparse mask $mask_B$, which can then be merged with the prior pruned task vector without any parameter overlap.

**Minimizing Overlap When Sparsity Limits are Exceeded**. When the sum of the sparsity levels across all task vectors exceeds 1 (e.g., when each vector retains 75% of its parameters), it becomes impossible to achieve zero overlap. In such cases, the objective shifts from eliminating overlap to minimizing it as much as possible. Additional pruning steps are applied selectively to reduce the extent of overlap between task vectors. For the detailed implementation of this process, please refer to Appendix A.3.

### 4.3 BALANCED SPARSIFICATION (BS)

**Motivation**. While CA can effectively reduce overlap, it does not address the imbalance in weight distribution that can arise within task vectors. These imbalances often lead to suboptimal performance in the merged model, affecting both its stability and efficiency. To mitigate this problem, we propose the Balanced Sparsification (BS) strategy, which enhances CA by addressing these imbalances and further improving the model's overall performance.

**Balanced Sparsification**. In BS, the weight matrix is divided into disjoint blocks of $m$ consecutive weights, and within each block, the $n$ weights with the largest absolute magnitude are retained, while the rest are pruned. This strategy is applied uniformly across all layers to ensure a more even weight distribution within each task vector. Minimizing imbalances prevents performance degradation of the merged models. For a more detailed discussion about the differences between Balanced Sparsification (BS) and n:m pruning, please refer to Appendix A.4.

We also discuss the flexibility and efficiency of the CABS framework in Appendix A.5. CABS can be integrated with other model merging techniques, where CA and BS can be applied independently or combined with other approaches to further enhance model merging. Additionally, our analysis of computational cost shows that CABS introduces virtually no additional overhead compared to standard merging methods, making it an efficient and adaptable solution for various model merging scenarios.

## 5 EXPERIMENTS

We conducted extensive experiments to evaluate the effectiveness of CABS in model merging. Our goal was to demonstrate that CABS can enhance both performance and stability across models of various scales, covering a diverse range of tasks.

### 5.1 EXPERIMENTAL SETUP

**Datasets and Models for Decoder-based Language Model Experiments.** For large-scale model evaluation, we utilized the LLM Leaderboard benchmark, encompassing six key tasks: AI2 Reasoning Challenge (Clark et al., 2018), HellaSwag (Zellers et al., 2019), MMLU (Hendrycks et al., 2020), TruthfulQA (Lin et al., 2022), Winogrande (Sakaguchi et al., 2021), and GSM8K (Cobbe et al., 2021). These tasks were assessed using the Eleuther AI Language Model Evaluation Harness (Gao et al., 2024), a standardized framework designed to test generative language models across various tasks. The decoder-based models used in our experiments were based on the Mistral-7b-v0.1 backbone and included fine-tuned variants such as WildMarcoroni-Variant1-7B and WestSeverus-7B-DPO-v2. For more detailed information on these datasets and models, please refer to Appendix A.6.

**Datasets and Models for Encoder-based Language Model Experiments.** For evaluating small-scale models, we utilized the GLUE benchmark, which includes a diverse set of tasks that can

be broadly categorized into four types: (1) acceptability judgments (e.g., CoLA (Warstadt et al., 2019)), (2) sentiment analysis (e.g., SST-2 (Socher et al., 2013)), (3) paraphrase detection (e.g., MRPC (Dolan & Brockett, 2005)), and (4) natural language inference (e.g., RTE (Dagan et al., 2005; Bar-Haim et al., 2006; Giampiccolo et al., 2007; Bentivogli et al., 2009)). Each of these tasks was chosen to represent a distinct aspect of natural language understanding for our model merging experiments. Due to the unavailability of test labels for GLUE, we utilized the original validation sets as test sets in our experiments. The models used for these tasks were pre-trained and fine-tuned versions of RoBERTa, obtained from Hugging Face. Further details regarding the models and tasks are provided in Appendix A.7.

**Evaluation Metrics.** Performance was evaluated primarily using accuracy for GLUE tasks, including CoLA, where MCC is typically recommended. This choice was made to maintain consistency across the GLUE benchmark and simplify averaging across tasks. For tasks from the LLM Leaderboard, we used task-specific metrics, such as success rates and accuracy, depending on the default evaluation metric for each task. Detailed explanations of the evaluation metrics and the rationale behind these choices can be found in Appendix A.8.

**Baselines.** We compared CABS against several baseline methods in two main categories: conflict handling and sparsification strategies. For conflict handling, we used Task Arithmetic (averaging task vectors) (Ilharco et al., 2022), TIES-Merging (pruning low-magnitude deltas and resolving sign conflicts) (Yadav et al., 2024), and our Conflict-Aware (CA) method, which sequentially prunes and masks overlapping weights. For sparsification, we compared DARE (random weight dropping with rescaling) (Yu et al., 2024), Magnitude Pruning (retaining highest-magnitude weights) (Zhu & Gupta, 2018), and our Balanced Sparsification (BS) method, which applies n:m pruning to balance weight distribution.

**Grid Search of Rescale Factor $\lambda$.** For small-scale tasks, we performed a fine-grained $\lambda$ parameter search with an interval of 0.01 (compared to the 0.1 used in previous works) to ensure fair comparisons across methods. In contrast, because of the high computational cost of large-scale experiments (e.g., with 7B models), we followed prior work by adopting a coarser grid interval of 0.1, with equal $\lambda$ values for all vectors. The impact of lambda grid intervals is discussed in Appendix A.9, showing how coarser intervals may lead to unfair comparisons by missing optimal values. Detailed steps for our grid search strategy are outlined in Appendix A.10.

**Implementation Details.** The model evaluations were performed on A100-40GB GPUs. For small-scale and discriminative tasks in GLUE, we conducted a single evaluation per model, as minimal variance was observed across repeated runs. In contrast, for generative tasks involving large models, where results can be more variable, inference was implemented via the lm-evaluation-harness v0.4.0. To ensure consistency and robustness, we performed three evaluations and reported the average outcome. As for the hyperparameters of generative LMs, we set the maximum generation token limit to 256, the temperature to 1.0 for sampling, and the maximum context length to 2048 tokens.

## 5.2 PERFORMANCE OF CABS ON ENCODER-BASED LMS

This experiment validates the effectiveness of CABS in merging small-scale encoder-based models, such as RoBERTa, on tasks from the GLUE benchmark. For example, we merge two models fine-tuned on RTE and MRPC tasks, respectively, using CABS and baseline methods.

Table 1 presents the accuracy achieved by each method. Among the baselines, "Task Arithmetic" serves as a vanilla approach without any pruning, while the other four baselines incorporate pruning. We observe that all the pruning-enhanced baselines outperform the vanilla version, with an improvement of up to 0.50 achieved by "TIES-Merging + DARE", highlighting the effectiveness of the pruning technique in model merging. Furthermore, the baselines enhanced by random pruning (i.e., "+ DARE") surpass those enhanced by magnitude pruning (i.e., "+ Magnitude" and "TIES-Merging"), indicating that magnitude pruning underperforms random pruning due to the issues we have identified (refer to Section 3). By addressing these issues, CABS achieves a significant performance improvement over all baselines.

CABS achieves a performance gain of 1.34 over "Task Arithmetic", which is **168%** greater than the improvement of 0.50 achieved by the SOTA baseline "TIES-Merging + DARE" Additionally, in normalized accuracy (shown in column "AVG-N"), CABS showed a relative improvement of **202%**

over the best-performing enhanced baseline (+1.57 vs. +0.52). For similar results on the CoLA and SST-2 tasks, please refer to Table 8 in the appendix A.11.

Table 1: Performance comparison on RTE-MRPC task pair using different methods (sparsity=0.9).

| METHOD | RTE | MRPC | AVG | RTE-N | MRPC-N | AVG-N |
|---|---|---|---|---|---|---|
| Fine-tuned model on RTE | 79.42 | 25.98 | 52.70 | 100.00 | 28.51 | 64.26 |
| Fine-tuned model on MRPC | 47.29 | 91.18 | 69.24 | 59.54 | 100.00 | 79.77 |
| Task Arithmetic | 73.29 | 87.01 | 80.15 | 92.23 | 95.42 | 93.82 |
| Task Arithmetic + Magnitude | **74.73** | 86.03 | 80.38(+0.23) | **94.12** | 94.35 | 94.24(+0.42) |
| Task Arithmetic + DARE | 72.92 | 88.24 | 80.58(+0.43) | 91.82 | 96.78 | 94.30(+0.48) |
| TIES-Merging | 74.37 | 86.03 | 80.20(+0.05) | 93.64 | 94.35 | 94.00(+0.18) |
| TIES-Merging + DARE | 72.56 | 88.73 | 80.65(+0.50) | 91.36 | 97.31 | 94.34(+0.52) |
| **CABS (Ours)** | 74.01 | **88.97** | **81.49(+1.34)** | 93.20 | **97.58** | **95.39(+1.57)** |

## 5.3 Performance of CABS on Decoder-based LMs

The "AVG" columns in Tables 2 and 3 present the average performance of each method across six tasks, demonstrating that CABS outperforms all baselines on generative LMs. Table 2 shows the results at a sparsity level of 0.25, where CABS can minimize the overlap, reaching an accuracy of 76.48%. In Table 3, at a sparsity level of 0.75, where CABS can eliminate overlap entirely, resulting in a performance improvement compared to the 0.25 sparsity level (76.50 vs. 76.48).

It is worth mentioning that, to figure out how far current model merging methods are from the expectation of the research field, we introduce an "ideal model" as a strict and meaningful baseline. The ideal model represents a hypothetical scenario where the merged model achieves optimal performance for each task, which is "constructed" by selecting the best-performing individual task-specific model for each task.

Table 2: Performance comparison on LLM Leaderboard using different methods (sparsity=0.25).

| METHOD | ARC | Hella. | MMLU | TQA | Wino. | GSM8K | AVG |
|---|---|---|---|---|---|---|---|
| WestSeverus-7B-DPO-v2 | 71.30 | 88.26 | 63.92 | 72.72 | 83.69 | 74.27 | 75.69 |
| WildMarcoroni-Variant1-7B | 73.63 | 88.67 | 63.96 | 70.07 | 84.34 | 74.48 | 75.86 |
| ideal model | 73.63 | 88.67 | 63.96 | 72.72 | 84.34 | 74.48 | 76.30 |
| Task Arithmetic(Dense) | 72.52 | 89.25 | 63.39 | 74.00 | 83.46 | 73.38 | 76.02(-0.28) |
| Task Arithmetic + Magnitude | 71.67 | 89.15 | 63.42 | 74.05 | 84.37 | 73.53 | 76.03(-0.27) |
| Task Arithmetic + DARE | 72.30 | 88.77 | **63.84** | 72.08 | 84.40 | 74.40 | 75.96(-0.34) |
| TIES-Merging | 72.41 | **89.34** | 63.40 | 74.03 | 83.64 | 73.69 | 76.09(-0.21) |
| TIES-Merging + DARE | 72.30 | 88.63 | 63.76 | 72.16 | 85.06 | 74.37 | 76.05(-0.25) |
| TIES-Merging + CABS | **72.97** | 89.20 | 63.46 | 74.00 | **85.16** | 74.50 | 76.44(+0.14) |
| **CABS (Ours)** | 72.75 | 89.17 | 63.48 | **74.08** | 84.66 | **74.73** | **76.48(+0.18)** |

In the "AVG" columns of Tables 2 and 3, the numbers in parentheses indicate the difference between the method's average accuracy and that of the ideal model. On the one hand, the outcome highlights a significant advantage of model merging: the enhancement of generalization. While the merged model may not surpass the ideal model on every individual task, it often achieves superior performance on specific tasks due to improved generalization capabilities. For example, in the TruthfulQA task (see column "TQA" in Table 3), the fine-tuned models scored 72.72 and 70.07, whereas the vanilla baseline reached 74.00, and CABS further boosted the score to 74.41. On the other hand, we can see, CABS achieved an average performance of 76.50, exceeding the ideal virtual model's performance of 76.30. In comparison, the highest-performing baseline scored 76.09, with a drop of 0.21 compared to the ideal model. The results demonstrate the effectiveness of CABS in enhancing model generalization and robustness. This success underscores the value of CABS for model merging in large-scale models.

Table 3: Performance comparison on LLM Leaderboard using different methods (sparsity=0.75).

| METHOD | ARC | Hella. | MMLU | TQA | Wino. | GSM8K | AVG |
|---|---|---|---|---|---|---|---|
| WestSeverus-7B-DPO-v2 | 71.30 | 88.26 | 63.92 | 72.72 | 83.69 | 74.27 | 75.69 |
| WildMarcoroni-Variant1-7B | 73.63 | 88.67 | 63.96 | 70.07 | 84.34 | 74.48 | 75.86 |
| ideal model | 73.63 | 88.67 | 63.96 | 72.72 | 84.34 | 74.48 | 76.30 |
| Task Arithmetic(Dense) | 72.52 | 89.25 | 63.39 | 74.00 | 83.46 | 73.38 | 76.02(-0.28) |
| Task Arithmetic + Magnitude | 71.93 | **89.32** | 63.18 | 73.85 | 84.12 | 72.22 | 75.77(-0.53) |
| Task Arithmetic + DARE | 72.64 | 88.86 | 63.54 | 72.82 | 84.03 | 73.44 | 75.89(-0.41) |
| TIES-Merging | 71.42 | 89.17 | 63.16 | 73.82 | **84.74** | 73.01 | 75.89(-0.41) |
| TIES-Merging + DARE | 71.87 | 88.95 | **63.56** | 72.87 | 84.61 | 73.21 | 75.85(-0.46) |
| **CABS (Ours)** | **72.92** | 88.89 | 63.50 | **74.41** | 84.63 | **74.65** | **76.50(+0.20)** |

## 5.4 ABLATION STUDIES AND DISCUSSION

Within the CABS framework, we first analyze the independent contributions of CA and BS by examining the impact of parameter overlap and unbalanced weight distribution on model merging. Next, we perform ablation studies to isolate the contributions of CA and BS, demonstrating the importance of both strategies for achieving optimal results.

**Performance Impact of Overlap Rate.** We examined the impact of varying overlap rates on final model performance to validate the importance of CA. The experiment was conducted on two task pairs (RTE-MRPC and CoLA-SST2) with a fixed sparsity level of 0.50, using random pruning for fair comparison. We first pruned one task vector, then adjusted the pruning of second vector by controlling the ratio of retained weights in the overlapping and non-overlapping regions to achieve target overlap rate, ranging from 0% (no overlap, CA) to 100% (full overlap).

Figure 4 shows that lower overlap generally leads to better performance, underscoring the importance of reducing parameter overlap, as achieved through CA. The 50% overlap point, which corresponds to the expected overlap rate of DARE, is noteworthy but does not perform as well as the no overlap condition (CA). This, along with the 0% and 100% overlap points, has been specifically highlighted in the figure for clarity.

CA becomes particularly critical at lower sparsity levels. For example, at 0.5 sparsity, the number and rate of overlapping parameters are much higher than at 0.9 sparsity. This makes CA especially valuable at lower sparsity levels, where task vectors retain more parameters and are thus more likely to result in significant overlap.

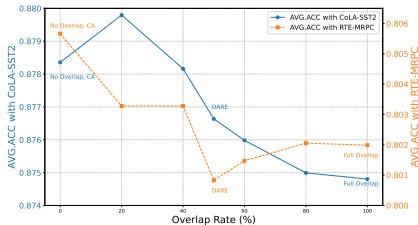

Figure 4: Merged model performance decreases as overlap rate increases, underscoring the importance of CA in reducing conflicts.

**Performance Impact of Balanced Sparsification.** Next, we evaluated BS's effectiveness by comparing different sparsity strategies, including layer-wise magnitude pruning, row-wise magnitude pruning, and n:m magnitude pruning. Table 4 presents the results at a sparsity level of 0.9, demonstrating that n:m magnitude pruning outperforms other methods, as it maintains balanced weight distribution and in turn improves model stability. BS proves most effective at a high sparsity level, such as 0.9, where the risk of unbalanced pruning is much higher. By ensuring that retained weights are distributed evenly across blocks, BS mitigates the potential for performance degradation due to concentrated weight distributions. As shown in Table 4, BS achieves an average performance of 81.30, outperforming layer-wise pruning (80.38) and row-wise pruning (80.61), and delivering a significant improvement over the base task-arithmetic score of 80.15. This highlights the crucial role of BS in enhancing model performance at high sparsity levels.

**Combined Effect of CA and BS.** To further explore the combined effect of CA and BS within CABS, we compared the full implementation of CABS with configurations that included only CA or BS. The results in Table 5 show that while CA and BS independently contribute to performance

Table 4: Comparison of sparsity strategies: layer-magnitude, row-magnitude, and BS (sparsity=0.9). "TA" means "Task Arithmetic".

| METHOD | RTE | MRPC | AVG | RTE-N | MRPC-N | AVG-N |
|---|---|---|---|---|---|---|
| Fine-tuned model on RTE | 79.42 | 25.98 | 52.70 | 100.00 | 28.51 | 64.26 |
| Fine-tuned model on MRPC | 47.29 | 91.18 | 69.24 | 59.54 | 100.00 | 79.77 |
| Task Arithmetic(Dense) | 73.29 | 87.01 | 80.15 | 92.23 | 95.42 | 93.82 |
| TA + Magnitude(layer-wise) | **74.73** | 86.03 | 80.38(+0.23) | **94.12** | 94.35 | 94.24(+0.42) |
| TA + Magnitude(row-wise) | 74.06 | 87.05 | 80.61(+0.46) | 93.25 | 95.47 | 94.36(+0.54) |
| **TA + BS (Ours)** | 74.37 | **88.23** | **81.30(+1.08)** | 93.64 | **96.76** | **95.20(+1.38)** |

improvements, their combination within CABS achieves the highest accuracy and stability across different sparsity levels.

Table 5: Ablation study of CABS across different sparsity levels.

| Sparsity Level | Method | Overlap Rate | Avg Accuracy |
|---|---|---|---|
| 0% | Task Arithmetic | 100.00 | 76.02 |
| 25% | TA-magnitude | 80.69 | 76.03 |
| | CA Only | 66.67 | 76.29 |
| | BS Only | 80.97 | 76.33 |
| | CABS | 66.67 | 76.48 |
| 75% | TA-magnitude | 71.42 | 75.77 |
| | CA Only | 0.00 | 76.21 |
| | BS Only | 58.63 | 76.24 |
| | CABS | 0.00 | 76.50 |

In conclusion, our ablation studies confirm the necessity of reducing overlap rates and maintaining balanced weight distribution for optimal model merging. They validate the crucial roles of CA and BS, showing that combining both strategies achieves the best performance across various tasks and sparsity settings.

Additionally, we performed a series of analyses on the impact of different sparse sequences, and varying n:m ratios. These results, which further elucidate the robustness of the CABS framework, are provided in the Appendix A.12 and A.13. We also conducted rescaling experiments and found that applying rescaling to magnitude-pruned task vectors can restore performance to levels comparable to the original models, similar to what has been observed with DARE's random pruning method. Detailed results of these rescale experiments are included in Appendix A.14.

## 6 CONCLUSION

In this work, we identified the issues of high parameter overlap and unbalanced weight distribution in task vector sparsification. We then proposed the CABS framework to address these challenges in model merging. CABS effectively reduces overlap and ensures a more balanced distribution of retained weights, resulting in improved performance across various tasks and model sizes. The CABS framework can be integrated into existing model merging techniques. Extensive experiments on both small- and large-scale models demonstrated CABS's effectiveness in improving model performance and maintaining model generalization. We also conducted a detailed analysis of CABS's components, providing insights into its robust handling of sparsification challenges in model merging. For a discussion on limitations and future work, see Appendix A.15.

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

# A APPENDIX

## A.1 OVERLAP RATE CALCULATION

The overlap rate between two task vectors is a metric used to quantify the extent to which the same parameters are retained after pruning. This metric is particularly useful in understanding how pruning strategies impact the sharing of model parameters across different tasks, which can lead to conflicts during model merging.

The overlap rate is calculated as follows: Given two task vectors $\tau_A$ and $\tau_B$, the overlap rate is defined as the ratio of the number of shared non-zero parameters to the total number of non-zero parameters in the first task vector $\tau_A$. Mathematically, this can be expressed as:

$$\text{Overlap Rate} = \frac{|\tau_A \cap \tau_B|}{|\tau_A|}$$

where $|\tau_A \cap \tau_B|$ represents the count of non-zero parameters that are common to both vectors $\tau_A$ and $\tau_B$, and $|\tau_A|$ denotes the total count of non-zero parameters in vector $\tau_A$. This calculation shows the extent of overlap between two task vectors. A higher overlap rate means more shared parameters, increasing the potential for conflicts during model merging.

## A.2 WEIGHT DISTRIBUTION ANALYSIS ACROSS LAYERS AND SPARSITY RATIOS

This section provides a comprehensive analysis of the heatmaps illustrating weight distributions across different layers of the model and various sparsity ratios. Figures 5-7 show the weight distribution for four representative layers: self_attn.k_proj.weight (layer 6), self_attn.q_proj.weight (layer 12), self_attn.v_proj.weight (layer 24), and mlp.up_proj.weight (layer 18) at sparsity ratios of 25%, 50%, 75%, and 90%.

These heatmaps demonstrate how increasing sparsity causes magnitude-based pruning to concentrate weights in localized regions of the parameter space. As the sparsity level increases, this clustering becomes more pronounced, especially at 75% and 90% sparsity levels, leading to potential imbalances that can degrade model performance.

The recurring pattern across all layers further highlights the significance of strategies like Balanced Sparsification (BS), which aim to distribute weights more evenly across the model. By ensuring a more uniform distribution of the retained weights, BS helps to maintain model stability and performance after sparsification.

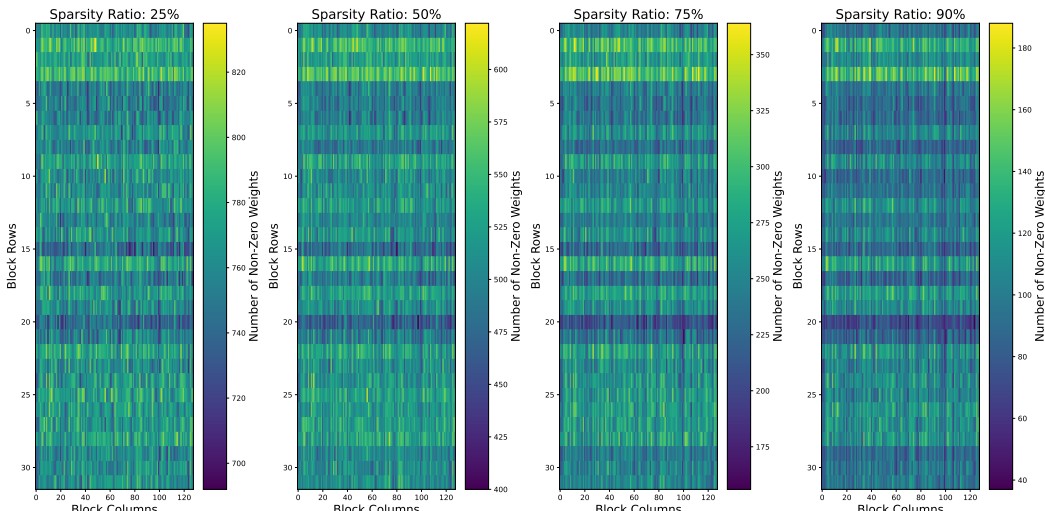

Figure 5: Heatmaps of weight distribution in model.layers.6.self_attn.k_proj.weight across different sparsity ratios (25%, 50%, 75%, and 90%).

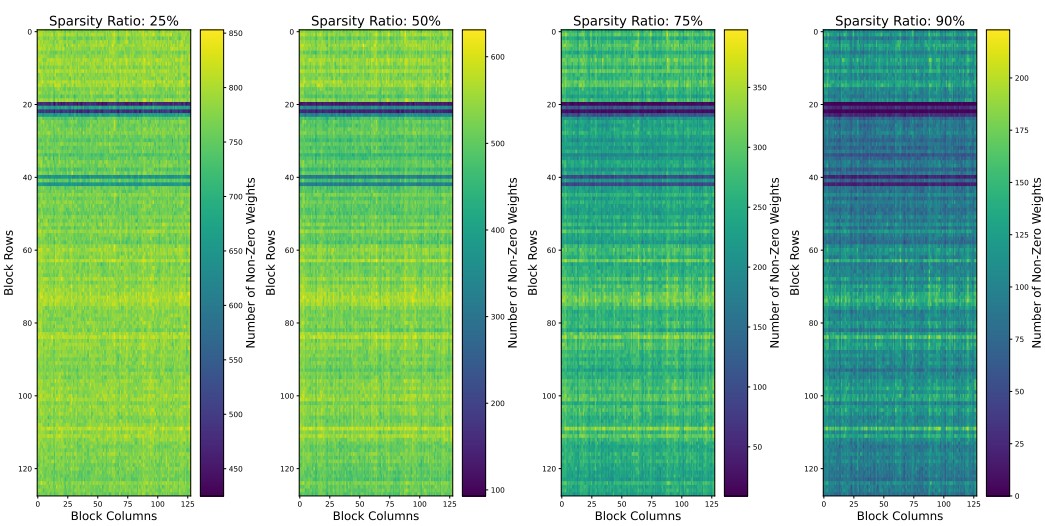

Figure 6: Heatmaps of weight distribution in model.layers.12.self_attn.q_proj.weight across different sparsity ratios (25%, 50%, 75%, and 90%).

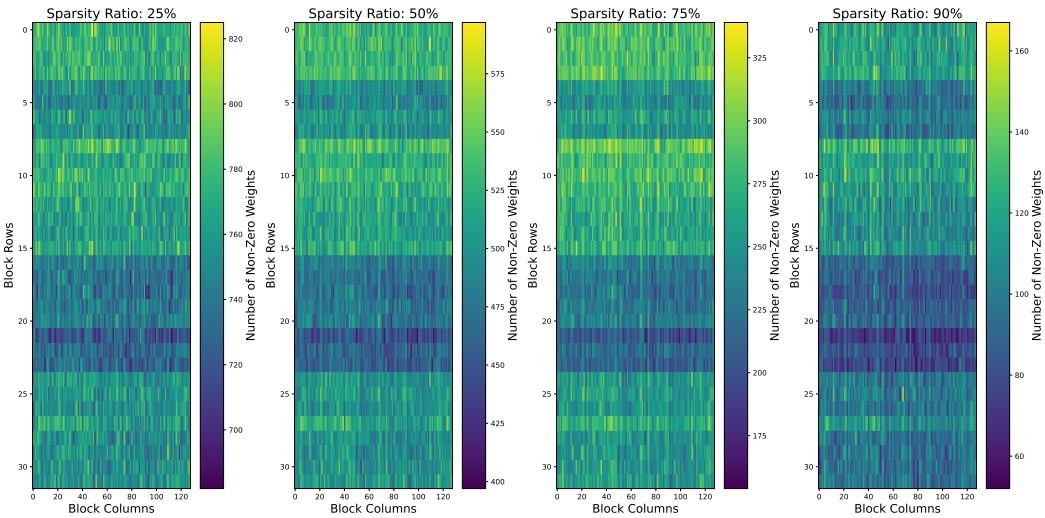

Figure 7: Heatmaps of weight distribution in model.layers.18.mlp.up_proj.weight across different sparsity ratios (25%, 50%, 75%, and 90%).

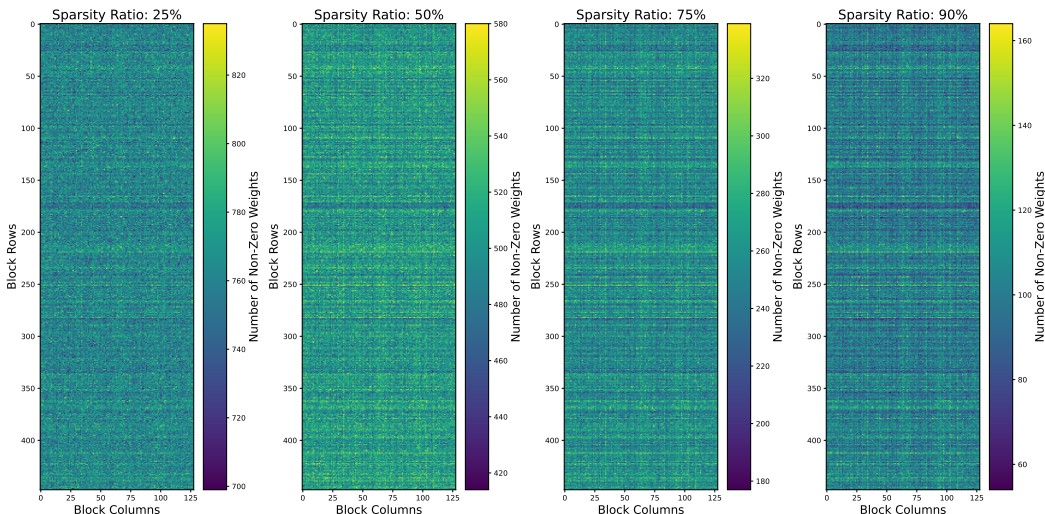

Figure 8: Heatmaps of weight distribution in model.layers.24.self_attn.v_proj.weight across different sparsity ratios (25%, 50%, 75%, and 90%).

### A.3    ALGORITHM OF LOW-OVERLAP SPARSITY

In this section, we provide the detailed algorithm for Low-Overlap Sparsity in Algorithm 2, designed to minimize direct conflicts during the model merging process. The algorithm sequentially applies sparsification to task vectors, ensuring that the non-overlapping portions of the task vectors are prioritized, thereby reducing coupling and conflict between different tasks in the final merged model.

---

**Algorithm 2** CABS Implementation:minimize overlap rate

---

**Input:** Task vectors $\tau_A$, $\tau_B$, base model $W_{\text{base}}$, sparsity level $n$ , $m$, rescale factors $\lambda_A$ , $\lambda_B$
**Output:** Merged model parameters $W_{\text{final}}$
  1: Apply n:m pruning to $\tau_A$ and compute $\text{mask}_A$                                       // include BS
  2: Compute initial_$\text{mask}_B = 1 - \text{mask}_A$, retaining non-overlapping regions of $\tau_B$
  3: If initial_$\text{mask}_B$ retains less than $n \div m$ of weights, update $\text{mask}_B$ by including additional weights from the overlapping region $\text{mask}_A \odot \tau_B$ until the target sparsity $n \div m$ is reached
  4: Merge the pruned vectors with the base model:

$$W_{\text{final}} = W_{\text{base}} + \lambda_A \times \text{mask}_A \odot \tau_A + \lambda_B \times \text{mask}_B \odot \tau_B$$

  5: **return**  $W_{\text{final}}$

---

### A.4    COMPARISON OF N:M PRUNING AND BS

Although both n:m pruning and BS employ the same operation—selecting the top $n$ values out of $m$ consecutive weights based on magnitude—their goals and use cases differ:

- *Goal*: The primary goal of n:m pruning is to achieve model compression and acceleration by reducing computational and memory costs. In contrast, BS is designed to maintain a balanced distribution of task vectors while minimizing conflicts between them during merging, not to merely discard unimportant weights.

- *Result*: n:m pruning is typically used for structured pruning in models, aiming to reduce inference time and memory usage. On the other hand, BS is applied specifically to task vectors. After the task vectors are merged with a base model, the resulting model remains dense, meaning that the practical computation and memory savings are not realized, but the model gains improved capacity.

- *Sparsity Ratios*: n:m pruning often uses configurations like 2:4 or 4:8, where the sparsity level is generally around 50%. In contrast, the sparsification of task vectors under BS can involve much higher sparsity levels, as can be seen in Table 11 (Appendix A.13), with configurations such as 64:256 at 75% sparsity.

- *Effectiveness*: Typically, n:m pruning yields lower performance compared to magnitude pruning in compression tasks, as the more strict uniform distribution of sparsity across blocks (e.g., every 4 weights) tends to hurt performance. However, in model merging, n:m sparsity can outperform row-wise or layer-wise magnitude pruning due to its more balanced distribution.

## A.5 Flexibility, Integration, and Efficiency

**Flexibility and Integration of CABS**. The CABS framework offers flexibility in its integration with existing model merging techniques. CA and BS can be applied independently or in combination with other approaches. For instance, CA can be combined with other sparsification strategies (e.g., DARE, magnitude) to minimize parameter overlap, while BS can implement n:m sparsity using different importance metrics beyond magnitude. Additionally, at lower sparsity levels, CABS can be effectively combined with techniques like Ties-merging to solve sign conflict, making it adaptable to various merging scenarios.

**Efficiency and Complexity**. Model merging, as implemented in toolkits like MergeKit (Goddard et al., 2024), inherently has low computational overhead since it bypasses full model retraining. In CABS, we introduce minimal additional cost to the merging process. The Conflict-Aware (CA) strategy modifies the pruning process from parallel to sequential, with the addition of a mask inversion and element-wise product to avoid overlap between task vectors. These operations introduce negligible computational overhead, especially given that most modern sparsification frameworks, including MergeKit, already adopt a sequential approach.

For Balanced Sparsification (BS), while extreme n:m pruning ratios (e.g., 32:128) may not benefit from hardware-level acceleration available for smaller ratios like 2:4, BS remains efficient in terms of time complexity. Here, $N$ represents the total number of parameters in a given layer of the model. Instead of performing a full sorting operation across the entire layer as in layer-wise magnitude pruning (which has a time complexity of $O(N \log N)$), BS operates by selecting $n$ weights within smaller fixed windows of size $m$. This process involves sorting each window of size $m$, resulting in a time complexity of $O(N \log m)$, which is more efficient than the global sorting required for layer-wise pruning.

In conclusion, while CABS introduces additional steps to improve weight distribution and mitigate overlap, these steps have minimal impact on the overall computational cost, ensuring that the merging process remains efficient.

## A.6 Details of Datasets and Models for Decoder-based LMs

**Tasks** The LLM Leaderboard benchmark consists of six primary tasks designed to evaluate the capabilities of large-scale generative language models across various domains:

- **AI2 Reasoning Challenge**: A set of grade-school science questions aimed at testing reasoning capabilities.

- **HellaSwag**: A commonsense inference test that is challenging for SOTA models but easy for humans ( 95% accuracy).

- **MMLU**: A multitask accuracy test covering 57 tasks, including elementary mathematics, US history, computer science, law, and more.

- **TruthfulQA**: A test measuring a model's propensity to reproduce falsehoods commonly found online.

- **Winogrande**: An adversarial and difficult Winograd schema-based benchmark for commonsense reasoning.

- **GSM8K**: A set of grade school math word problems designed to measure a model's ability to solve multi-step mathematical reasoning problems.

**Models** The decoder-based models used in our evaluations were built upon the Mistral-7b-v0.1[1] backbone and included several fine-tuned variants: WildMarcoroni-Variant1-7B[2],WestSeverus-7B-DPO-v2[3]

These models were chosen for their ability to perform well across the diverse set of tasks included in the LLM Leaderboard benchmark and their use in prior research.

## A.7 DETAILS OF DATASETS AND MODELS FOR ENCODER-BASED LMS

**Tasks** The GLUE benchmark includes a variety of tasks designed to evaluate different aspects of natural language understanding. For our experiments, we selected the following four tasks:

- CoLA (Corpus of Linguistic Acceptability), which evaluates the grammatical acceptability of sentences with performance measured using the Matthews Correlation Coefficient (MCC);

- SST-2 (Stanford Sentiment Treebank), a binary sentiment classification task assessing whether a sentence expresses a positive or negative sentiment, evaluated using accuracy;

- MRPC (Microsoft Research Paraphrase Corpus), a paraphrase identification task where models predict whether two sentences have the same meaning, evaluated using both accuracy and F1 score;

- RTE (Recognizing Textual Entailment), a natural language inference task where models determine whether a hypothesis is true based on a given premise, evaluated using accuracy.

**Models** For each task, we utilized pre-trained and fine-tuned versions of RoBERTa, obtained from Hugging Face. Specifically, we used FacebookAI/roberta-base[4] as base model. textattack/roberta-base-CoLA [5], textattack/roberta-base-SST-2[6], textattack/roberta-base-MRPC[7], and textattack/roberta-base-RTE[8].

## A.8 EVALUATION METRICS

For GLUE tasks, accuracy was chosen as the uniform metric to facilitate fair comparison across tasks. While MCC is recommended for CoLA, we used accuracy to maintain consistency with other tasks. MCC typically reaches around 0.64 after fine-tuning for CoLA, whereas accuracy for other tasks often exceeds 0.9. This discrepancy makes it difficult to include MCC in an overall performance average.

For LLM Leaderboard tasks, the following metrics were used:

- **ARC**: Success rate (25-shot)

- **HellaSwag**: Accuracy (10-shot)

- **MMLU and Winogrande**: Accuracy (5-shot)

- **TruthfulQA**: Factual accuracy (0-shot)

- **GSM8K**: Success rate (5-shot)

These metrics provide a consistent and comparable basis for evaluating model performance across various benchmarks.

---

[1] https://huggingface.co/mistral-7b-v0.1
[2] https://huggingface.co/WildMarcoroni-Variant1-7B
[3] https://huggingface.co/WestSeverus-7B-DPO-v2
[4] https://huggingface.co/FacebookAI/roberta-base
[5] https://huggingface.co/textattack/roberta-base-CoLA
[6] https://huggingface.co/textattack/roberta-base-SST-2
[7] https://huggingface.co/textattack/roberta-base-MRPC
[8] https://huggingface.co/textattack/roberta-base-RTE

A.9 Impact of Lambda Search Grid on Performance

In this section, we analyze the impact of different lambda search grids on the performance of various model merging methods. Our experiments demonstrate the importance of using fine-grained grid intervals to fairly compare the effectiveness of these methods. Table 6 provides results across different grid intervals (0.01, 0.05, and 0.1) for several methods.

For most methods, performance declines as the grid interval increases, underscoring the importance of finer grids to accurately capture optimal lambda values. Coarser grids often miss these values, leading to noticeable drops in performance.

Interestingly, the DARE method maintains stable performance even with coarser grids (0.05 and 0.1). This is because the optimal lambda for DARE happens to coincide with a multiple of 0.1, resulting in no significant performance loss with coarser grids. However, when we exclude such co-incidental "sweet spot" lambdas, as shown in Table 7, DARE also exhibits a significant performance drop. This observation reinforces the idea that fine grid intervals are crucial for a fair and thorough evaluation of all methods. A finer grid ensures that all methods have an equal opportunity to find the best-performing lambda, though this must be balanced with computational cost

On the other hand, the CABS method demonstrates robust performance across all grid intervals. It consistently outperforms other methods, and its relative insensitivity to grid coarseness suggests that CABS is more robust and reliable under varying hyperparameter settings. This robustness, combined with its superior performance, makes CABS a strong choice for model merging.

Table 6: Performance comparison across different lambda grid intervals."TA" means "Task Arithmetic"

| Grid Interval | Task Arithmetic | DARE | TA-Magnitude | TIES-DARE | TIES-Merging | CABS |
|---|---|---|---|---|---|---|
| 0.01 | 80.15 | 80.58(+0.43) | 80.38(+0.23) | 80.65(+0.40) | 80.20(+0.05) | **81.49(+0.91)** |
| 0.05 | 79.85 | 80.58(+0.73) | 79.90(+0.05) | 79.91(+0.06) | 79.84(-0.01) | **81.19(+1.34)** |
| 0.10 | 79.43 | 80.58(+1.15) | 79.66(+0.23) | 79.14(-0.29) | 79.83(+0.40) | **80.82(+1.39)** |

Table 7: Performance comparison across different lambda grid intervals excluding one pair sweet spot lambdas in DARE.

| Grid Interval | Task Arithmetic | DARE | TA-Magnitude | TIES-DARE | TIES-Merging | CABS |
|---|---|---|---|---|---|---|
| 0.01 | 80.15 | 80.58(+0.43) | 80.38(+0.23) | 80.65(+0.40) | 80.20(+0.05) | **81.49(+0.91)** |
| 0.05 | 79.85 | 79.44(-0.41) | 79.90(+0.05) | 79.91(+0.06) | 79.84(-0.01) | **81.19(+1.34)** |
| 0.10 | 79.43 | 78.55(-0.88) | 79.66(+0.23) | 79.14(-0.29) | 79.83(+0.40) | **80.82(+1.39)** |

A.10 Grid Search Details for Small-Scale Experiments

In our small-scale experiments, we employed a two-step grid search strategy to determine the optimal rescale factor $\lambda$ that maximizes average performance across multiple tasks.

**Grid Search Strategy** As the sparsity level increases, the range of potential optimal $\lambda$ values broadens, and performance typically follows a pattern of increasing and then decreasing with respect to $\lambda$. To address this, we first performed a manual search with a 0.1 interval, identifying the broader region where the optimal $\lambda$ is likely to reside. Based on the results of this initial search, we conducted a more fine-grained search using a 0.01 interval, focusing on the region identified in the first step.

Unlike a fixed-range search, this adaptive approach allowed us to zero in on the most effective rescale factors for each sparsity level, ensuring more precise performance optimization. The performance values presented in the main text correspond to the optimal $\lambda$ found through this two-step process.

## A.11 Additional Experiments on other Task Pairs for Small-Scale Experiments

In this section, we present additional results for the CoLA-SST2 task pair to complement the main text's findings on RTE and MRPC. These tasks were selected to further validate the robustness and effectiveness of the proposed **CABS** method across different types of natural language processing tasks, particularly focusing on tasks involving linguistic acceptability and sentiment analysis.

Table 8 provides a detailed comparison of various model merging methods on the CoLA and SST2 tasks. The **CABS** method demonstrates superior performance, achieving the highest average scores across both tasks. The normalized accuracy scores (COLA-N and SST2-N) further emphasize the effectiveness of the **CABS** method, showing consistent improvements over the baseline methods.

The modest gains observed in the CoLA-SST2 experiments, similar to those in the RTE-MRPC pair, can be attributed to the fine-grained lambda grid search. This search process, which fine-tunes the sparsification parameters, improves the overall performance across all methods, thereby reducing the performance gaps. However, **CABS** still outperforms other methods, indicating its robustness in handling task-specific nuances during model merging.

Table 8: Performance comparison on COLA-SST2 task pair using different methods.(sparsity=0.9)

| METHOD | COLA | SST2 | AVG | COLA-N | SST2-N | AVG-N |
|---|---|---|---|---|---|---|
| Fine-tuned model on COLA | 85.04 | 50.92 | 67.98 | 100.00 | 54.15 | 77.08 |
| Fine-tuned model on SST2 | 68.74 | 94.04 | 81.39 | 80.83 | 100.00 | 90.32 |
| Task Arithmetic | 81.59 | 92.89 | 87.24 | 95.94 | 98.78 | 97.36 |
| Task Arithmetic + Magnitude | 81.69 | 93.46 | 87.58(+0.34) | 96.06 | 99.38 | 97.72(+0.36) |
| Task Arithmetic + DARE | 81.78 | 93.46 | 87.62(+0.38) | 96.17 | 99.38 | 97.78(+0.42) |
| TIES-Merging | 81.21 | 93.58 | 87.40(+0.16) | 95.5 | 99.51 | 97.51(+0.19) |
| TIES-Merging + DARE | 81.78 | **93.69** | 87.74(+0.50) | 96.17 | **99.63** | 97.90(+0.54) |
| **CABS (Ours)** | **82.55** | 93.35 | **87.95(+0.71)** | **97.07** | 99.27 | **98.17(+0.81)** |

The results from these additional experiments support the conclusions drawn in the main paper, highlighting **CABS** as a robust and effective model merging technique across various tasks and evaluation metrics.

## A.12 Performance Impact of Sparsification Sequence

We analyze how different sparse sequences, referring to the order in which source models (e.g., "wild" and "west") undergo sparsification during the merging process, affect the merged model's performance. In this context, "wild-first" and "west-first" indicate which model is sparsified first. Our findings, summarized in Table 9, suggest that while the order of sparsification has some impact, the effect remains relatively small.

In our additional experiments merging four models (shown in Table 10), we further analyzed the effect of different pruning orders (e.g., CSRM, SCMR, RCMS, MRSC). While individual task performance showed slight variations, the overall average remained robust, ranging narrowly from **81.6375 to 81.7**. This demonstrates that CABS effectively handles pruning sequence variations while maintaining high average performance.

Issues caused by pruning order could potentially be addressed by techniques such as using adaptive lambda to give greater importance to later models or adopting variable sparsity ratios to better balance model contributions. However, given the minor impact of pruning order on overall performance, we currently consider this less critical for our method's practical application.

## A.13 Effect of Different n:m Ratios at Fixed Sparsity Levels

This section examines how different n:m ratios impact the performance of the merged model while keeping the overall sparsity fixed at 75%. The results in Table 11 indicate that while higher n:m

Table 9: Performance comparison across different sparse sequences on LLM Leaderboard taskss-parsity=75%

| METHOD | ARC | Hella. | MMLU | TQA | Wino. | GSM8K | AVG |
|---|---|---|---|---|---|---|---|
| WestSeverus-7B-DPO-v2 | 71.30 | 88.26 | 63.92 | 72.72 | 83.69 | 74.27 | 75.69 |
| WildMarcoroni-Variant1-7B | 73.63 | 88.67 | 63.96 | 70.07 | 84.34 | 74.48 | 75.86 |
| Ideal Model | 73.63 | 88.67 | 63.96 | 72.72 | 84.34 | 74.48 | 76.30 |
| Task Arithmetic(Dense) | 72.52 | 89.25 | 63.39 | 74.00 | 83.46 | 73.38 | 76.02(-0.28) |
| CABS(16:64)-wild-first | 72.30 | 88.87 | 63.47 | 74.27 | 84.77 | 74.12 | 76.3(+0.0) |
| CABS(16:64)-west-first | 72.44 | 89.08 | 63.11 | 73.38 | 84.79 | **75.11** | 76.32(+0.02) |
| CABS(32:128)-wild-first | 72.92 | 88.89 | **63.50** | 74.41 | 84.63 | 74.65 | **76.50(+0.20)** |
| CABS(32:128)-west-first | 72.58 | 89.19 | 63.19 | 74.22 | 85.16 | 74.15 | 76.42(+0.12) |
| CABS(64:256)-wild-first | **72.87** | 89.02 | 63.43 | **74.61** | 84.37 | 73.92 | 76.37(+0.07) |
| CABS(64:256)-west-first | 72.38 | **89.29** | 63.15 | 73.47 | **85.40** | 74.65 | 76.39(+0.09) |

Table 10: Performance comparison under different sparsification sequences (sparsity=0.9).

| METHOD | COLA | SST2 | RTE | MRPC | AVG |
|---|---|---|---|---|---|
| Task Arithmetic | 76.32 | 90.83 | 69.68 | 81.37 | 79.55 |
| CABS (CSRM) | 78.24 | 92.32 | 74.37 | 81.62 | 81.6375 (+2.09) |
| CABS (SCMR) | 78.52 | 91.97 | 73.65 | 82.60 | 81.685 (+2.14) |
| CABS (RCMS) | 77.76 | 92.09 | 75.09 | 81.62 | 81.64 (+2.09) |
| CABS (MRSC) | 76.89 | 92.09 | 74.73 | 83.09 | 81.7 (+2.15) |

ratios (e.g., 64:256) tend to show slight improvements, the overall impact of varying n:m ratios remains relatively subtle, suggesting that model performance is not highly sensitive to these values.

Table 11: Impact of different n:m ratios at 75% sparsity on LLM Leaderboard tasks

| METHOD | ARC | Hella. | MMLU | TQA | Wino. | GSM8K | AVG |
|---|---|---|---|---|---|---|---|
| WestSeverus-7B-DPO-v2 | 71.30 | 88.26 | 63.92 | 72.72 | 83.69 | 74.27 | 75.69 |
| WildMarcoroni-Variant1-7B | 73.63 | 88.67 | 63.96 | 70.07 | 84.34 | 74.48 | 75.86 |
| Ideal Model | 73.63 | 88.67 | 63.96 | 72.72 | 84.34 | 74.48 | 76.30 |
| Task Arithmetic(Dense) | 72.52 | 89.25 | 63.39 | 74.00 | 83.46 | 73.38 | 76.02(-0.28) |
| CABS(16:64) | 72.44 | 89.08 | 63.11 | 73.38 | 84.79 | **75.11** | 76.32(+0.02) |
| CABS(32:128) | **72.92** | 88.89 | **63.50** | **74.41** | 84.63 | 74.65 | **76.50(+0.20)** |
| CABS(64:256) | 72.38 | **89.29** | 63.15 | 73.47 | **85.40** | 74.65 | 76.39(+0.09) |

## A.14 RESCALE EXPERIMENTS

In previous research, TIES utilized magnitude pruning to reduce conflicts during task vector merging but did not include a rescale step. Subsequent work on DARE introduced a two-step process: random pruning followed by rescaling with a factor of $\frac{1}{1-p}$, where $p$ is the sparsity rate. DARE demonstrated that random pruning, when combined with rescaling, could restore performance to levels comparable to the original fine-tuned models. However, DARE did not explore the effect of rescaling on magnitude-pruned task vectors.

In our experiments, we evaluated the impact of rescaling on both magnitude-based and random pruning methods across different sparsity levels. As shown in Figure 9, rescaling allows magnitude-pruned task vectors to recover performance similar to that achieved by DARE, suggesting that rescaling is a crucial step for maintaining model performance post-pruning.

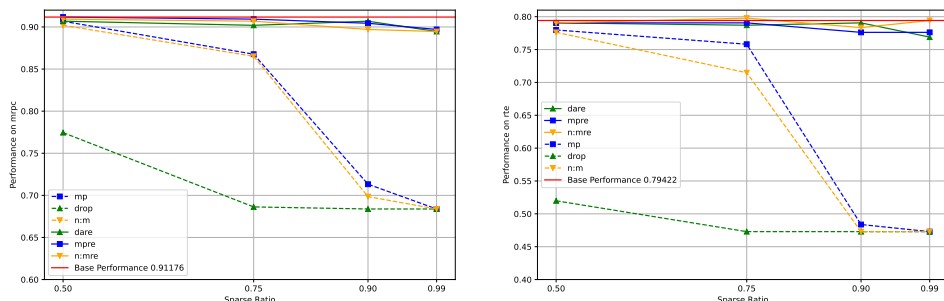

Figure 9: Impact of rescaling on different pruning methods across various sparsity levels. Performance is evaluated on RTE and MRPC tasks using RoBERTa. The horizontal axis represents the sparsity ratio, while the vertical axis indicates the performance of the task vectors after rescaling.

These findings confirm that, with appropriate rescaling, both magnitude-based and random pruning methods can achieve near-original performance. This insight complements the primary contributions of our work by showing that magnitude pruning, which traditionally underperformed compared to random pruning in TIES, can be equally effective when combined with rescaling. Although this experiment supports the robustness of magnitude pruning under rescale conditions, it is not the main focus of our study and is therefore detailed here in the appendix.

### A.15 Limitations and Future Work

**General Limitations.** Like other task vector-based methods, our approach is limited to models with identical architectures due to the element-wise operations used in merging model weights. This constraint restricts the generalization of the framework to models with homogeneous structures. Additionally, the reliance on manual tuning of the parameter $\lambda$ remains a common challenge, especially for large language models, requiring trial and error to optimize model performance.

**Limitations Specific to CABS.** CABS introduces two new hyperparameters—the sparse sequence and the n:m ratios—unique to its design, as discussed in Appendix A.12 and A.13. While these hyperparameters were not particularly sensitive in our experiments, they add complexity and increase computational cost. Furthermore, while CA and BS improve performance across various tasks, their effectiveness is reduced in scenarios where task vectors have minimal overlap or where models exhibit significant weight imbalances prior to sparsification. Additional experiments, especially at extreme sparsity levels or with heavily imbalanced models, are necessary to better understand these limitations.

**Future Work.** Several directions could help overcome these limitations. Expanding model merging techniques to include heterogeneous architectures or models trained from scratch represents a key area for future research. Additionally, improving the performance of merged models in multi-task settings—where current approaches do not yet match the performance of original single-task models—remains a priority. Automating the search for optimal hyperparameters, particularly $\lambda$, would reduce complexity and improve usability, especially in large-scale applications.

### A.16 Impact of Lambda on Performance

Figure 10 provides the average performance as a function of $\lambda$. It can be observed that within a certain range, the performance is relatively insensitive to variations in $\lambda$. This result corresponds to the performance of the CABS framework on the RTE-MRPC task. For visualization purposes, the same $\lambda$ values were used across the tasks rather than the task-specific $\lambda$ values reported in the paper. The $\lambda$ values range from 1 to 3, with a step size of 0.01, resulting in a total of 200 samples.

### A.17 Detailed Descriptions of Additional Model Merging Approaches

Evolutionary Model Merge (Akiba et al., 2024): This approach inspired by evolutionary algorithms that aims to improve large language models (LLMs) by simulating natural selection. It treats model

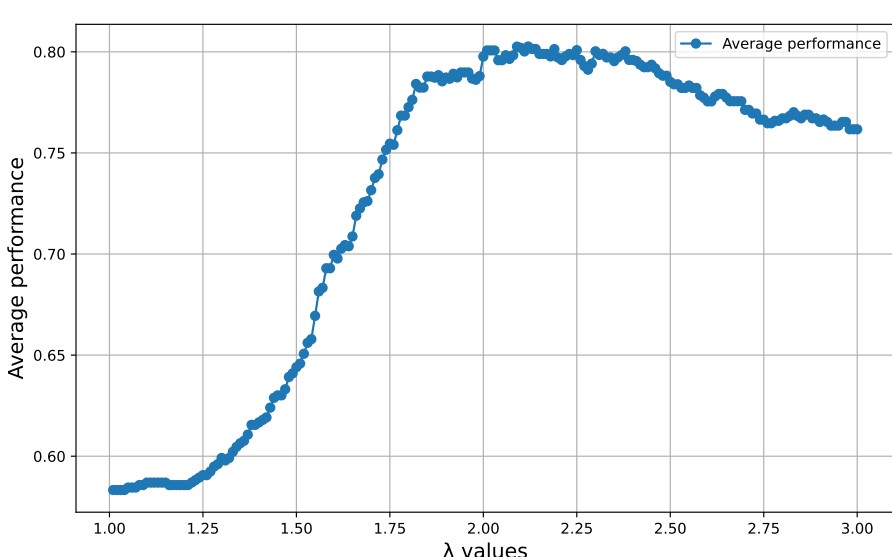

Figure 10: Average performance vs.lambda

merging as weight crossover and fine-tuning as weight mutation, iteratively refining models to create a single superior LLM. While EvoMerge has the potential to surpass traditional fine-tuning, it requires significant computational resources and incurs high merging overhead, limiting its practicality in resource-constrained settings.

Pack of LLMs (Mavromatis et al., 2024): Inspired by ensemble learning, this method combines multiple large language models into a unified framework, It assigns weights to LLMs based on perplexity scores, with two main approaches: a simple perplexity-weighted ensemble (PackLLMsim) and a more advanced greedy optimization method (PackLLMopt). While PackLLM is flexible and avoids additional training, it requires parallel storage and inference of multiple models, which can be resource-intensive and less suitable for latency-sensitive applications.

### A.18 EXTENDED EXPERIMENTS: MERGING MULTIPLE TASK VECTORS

To evaluate the method's ability to merge multiple task vectors (¿3), we conducted additional experiments merging four models at 90% sparsity. The same $\lambda$ value was used for each task vector, with a search interval of 0.01. This unified $\lambda$ approach simplifies the process and mitigates the computational burden of searching for optimal $\lambda$ combinations, which grows exponentially as the number of models $k$ increases.

As shown in Table 12, our method, **CABS**, achieves the highest average score of **81.7**, with a **+2.15 improvement** compared to the baseline task-arithmetic method. It also outperforms other methods, such as TIES-DARE, which achieves **79.8825 (+0.33)**, the best result among comparison methods.

The results highlight the scalability and effectiveness of CABS in merging multiple task vectors. CABS resolves conflicts and maintains balanced sparsity, achieving superior performance even under a unified $\lambda$ setting, which simplifies the merging process without compromising results.

### A.19 BALANCED SPARSITY (BS): UNIQUE ADVANTAGES IN MODEL MERGING

Balanced Sparsity (BS) distinguishes itself from other advanced pruning techniques through its primary objective: maintaining a balanced distribution of task vectors to minimize conflicts during model merging. In contrast, traditional pruning methods, such as SparseGPT (Frantar & Alistarh, 2023) and WANDA (Sun et al., 2023), focus on identifying and removing "unimportant" weights to

Table 12: Performance comparison of merging four task vectors at 90% sparsity.

| METHOD | COLA | SST2 | RTE | MRPC | AVG |
|---|---|---|---|---|---|
| Fine-tuned on COLA | 85.04 | 50.92 | 45.85 | 67.40 | 62.30 |
| Fine-tuned on SST2 | 68.74 | 94.04 | 46.93 | 68.38 | 69.52 |
| Fine-tuned on RTE | 31.83 | 51.15 | 79.42 | 25.98 | 47.10 |
| Fine-tuned on MRPC | 49.47 | 51.15 | 47.29 | 91.18 | 59.77 |
| Task Arithmetic | 76.32 | 90.83 | 69.68 | 81.37 | 79.55 |
| DARE | 76.99 | 90.14 | 70.76 | 81.13 | 79.76(+0.21) |
| TIES-DARE | 77.66 | 90.94 | 69.31 | 81.62 | 79.88(+0.33) |
| Magnitude | 82.07 | 87.04 | 65.34 | 79.66 | 78.53(-1.02) |
| TIES-Magnitude | 82.36 | 86.93 | 61.01 | 79.41 | 77.43(-2.12) |
| **CABS (Ours)** | **76.89** | **92.09** | **74.73** | **83.09** | **81.70(+2.15)** |

reduce redundancy. Ironically, these methods, while effective in preserving task-specific knowledge, often exacerbate conflicts during model merging by amplifying task vector interference.

Our experiments confirm that BS outperforms traditional pruning techniques in model merging tasks, as shown in Table 13. For instance, while SparseGPT and WANDA yield comparable performance to magnitude pruning, they fail to address the critical need for balanced task vector allocation. BS achieves an average score of 81.30 (+1.15 over Task Arithmetic with magnitude pruning), significantly outperforming SparseGPT (80.34) and WANDA (80.40). Furthermore, our method, CABS, which builds on BS, further improves the average performance to 81.49 (+1.34).

The fundamental difference lies in the purpose of task vector sparsification. Traditional pruning methods aim to optimize task-specific knowledge retention and often perform better under relaxed sparsity constraints (e.g., layer-wise 50% sparsity). However, in model merging, stricter sparsity requirements are beneficial, as demonstrated in Table 4. For example, layer-wise sparsification performs worse than row-wise sparsification, which is further outperformed by BS. This underscores the unique advantage of BS in prioritizing balance over redundancy reduction, a critical factor for merging performance.

Additionally, as discussed in Appendix A.14, task vector knowledge is inherently distributed across substructures, making it less sensitive to weight removal. Even random sparsification achieves near-full recovery of fine-tuning performance on target datasets, while random pruning fails entirely in traditional one-shot pruning scenarios. This highlights the robustness of task vector sparsification in preserving knowledge for merging tasks.

In conclusion, BS provides a unique solution to the challenges of task vector sparsification by maintaining balance and minimizing conflicts during merging. While traditional pruning techniques excel in redundancy reduction for single-task scenarios, they are less suited for the complexities of multi-task model merging. BS, tailored specifically for this purpose, delivers superior results and is critical to the success of our proposed method.

Table 13: Performance comparison of different pruning methods in model merging tasks.

| Method | RTE | MRPC | Avg |
|---|---|---|---|
| Task Arithmetic (Dense) | 73.29 | 87.01 | 80.15 |
| TA + Magnitude | 74.73 | 86.03 | 80.38 (+0.23) |
| TA + SparseGPT | 72.92 | 87.75 | 80.34 (+0.19) |
| TA + WANDA | 73.29 | 87.50 | 80.40 (+0.25) |
| TA + BS (Ours) | 74.37 | 88.23 | 81.30 (+1.15) |
| **CABS (Ours)** | **74.01** | **88.97** | **81.49 (+1.34)** |

## A.20 ADDITIONAL EXPERIMENTS ON GPT-2-BASED MODELS

Following your suggestion, we have extended our experiments to include other architectures, specifically GPT-2-based models (Radford et al., 2019). The results, summarized in Table 14, highlight the performance of CABS and other methods on tasks derived from **FusionBench** (Tang et al., 2024).

Table 14: Performance comparison on GPT-2-based models.

| Method | CoLA | MRPC | AVG |
|---|---|---|---|
| Fine-tuned on CoLA | 76.80 | 68.40 | 72.60 |
| Fine-tuned on MRPC | 30.80 | 80.39 | 55.60 |
| Ideal Model | 76.80 | 80.39 | 78.60 |
| Task Arithmetic (Dense) | 75.55 | 77.45 | 76.50 (-2.10) |
| TA + DARE | 76.70 | 77.21 | 76.95 (-1.65) |
| TA + Magnitude | 76.61 | 79.66 | 78.13 (-0.47) |
| TIES + DARE | 77.09 | 76.72 | 76.91 (-1.69) |
| TIES + Magnitude | 76.89 | 77.94 | 77.42 (-1.18) |
| **CABS (Ours)** | **76.41** | **80.88** | **78.65 (+0.05)** |

The results demonstrate that **CABS** outperforms all other methods and is the only method to surpass the Ideal Model. Although the improvement margin is relatively smaller due to the upper-bound constraint imposed by the Ideal Model, CABS consistently proves its effectiveness across tasks.

Interestingly, magnitude pruning shows unexpectedly strong results on GPT-2-based models, surpassing DARE by a significant margin. This contrasts with previous experiments on other architectures, suggesting a potential architecture-specific behavior in existing pruning methods. Nevertheless, CABS maintains its advantages across different architectures, showcasing its robustness and adaptability.

These findings underscore the versatility of CABS and its potential for diverse architectures.

## A.21 MULTILINGUAL APPLICABILITY OF CABS

While our primary experiments focused on English tasks to maintain comparability with prior work, we extended our evaluation to include two Korean language tasks, **kobest_copa** and **kobest_boolq** (Jang et al., 2022), to investigate the multilingual applicability of our method. These additional experiments provide insight into the performance of CABS across diverse linguistic contexts. The results are summarized in Table 15.

Table 15: Performance comparison on multilingual tasks, including Korean language benchmarks.

| Model | ARC | Hella. | MMLU | TQA | Wino. | GSM8K | Kcopa | Kboolq | Avg |
|---|---|---|---|---|---|---|---|---|---|
| Mistral-7B-v0.1 | 59.98 | 83.31 | 64.16 | 42.15 | 78.37 | 37.83 | 59.00 | 62.61 | 60.93 |
| WestSeverus | 71.30 | 88.26 | 63.92 | 72.72 | 83.69 | 74.27 | 63.30 | 81.91 | 74.92 |
| WildMarcoroni | 73.63 | 88.67 | 63.96 | 70.07 | 84.34 | 74.48 | 64.80 | 82.08 | 75.25 |
| Ideal Model | 73.63 | 88.67 | 63.96 | 72.72 | 84.34 | 74.48 | 64.80 | 82.08 | 75.59 |
| TA (Dense) | 72.52 | 89.25 | 63.39 | 74.00 | 83.46 | 73.38 | 65.60 | 72.58 | 74.27 (-1.32) |
| TA + Magnitude | 71.93 | 89.32 | 63.18 | 73.85 | 84.12 | 72.22 | 64.70 | 72.86 | 74.02 (-1.57) |
| TA + DARE | 72.64 | 88.86 | 64.53 | 72.82 | 84.03 | 73.44 | 61.40 | 79.34 | 74.63 (-0.96) |
| TIES-Merging | 71.42 | 89.17 | 63.16 | 73.82 | 84.74 | 73.01 | 64.80 | 73.08 | 74.15 (-1.44) |
| TIES + DARE | 71.87 | 88.95 | 63.56 | 72.87 | 84.61 | 73.21 | 61.40 | 79.63 | 74.51 (-1.08) |
| **CABS (Ours)** | **72.92** | **88.89** | **63.50** | **74.41** | **84.63** | **74.65** | **65.10** | **79.20** | **75.41 (-0.18)** |

For these experiments, we reused the merging configuration from our previous 7B experiments to ensure consistency across evaluations and to reduce computational overhead during this phase. CABS achieves an average score of **75.41**, closely matching the ideal model's performance of **75.59**

(a difference of -0.18). In comparison, the best alternative, Task Arithmetic + DARE, achieves **74.63** (-0.96), with other methods falling even further behind. These results confirm that CABS delivers competitive performance across both English and non-English tasks.

Additionally, these findings underscore the robustness of CABS in maintaining performance across multilingual benchmarks, highlighting its potential applicability to a wide of languages and tasks. While the absolute improvement margins may vary due to upper-bound constraints imposed by the ideal model, CABS consistently demonstrates its effectiveness and adaptability across diverse settings.

## A.22 PERFORMANCE AT 90% SPARSITY

To address concerns regarding performance under extreme sparsity levels, we conducted additional experiments at 90% sparsity for smaller models. As shown in Table 16, all methods experienced performance degradation due to the removal of a large number of parameters. Among the methods, TA-dare showed the most significant decline, likely due to the excessive pruning of critical parameters, leading to a drop of -3.08 in average score.

In contrast, our CABS approach demonstrated superior robustness, achieving the best performance across all methods with an average score of **76.10**. Notably, CABS outperformed Task Arithmetic(Dense) (**76.00**), further validating its generalization capabilities. These findings highlight CABS's ability to maintain competitive performance even under extreme sparsity conditions.

Based on these results, sparsity levels beyond 90% would likely lead to further performance degradation across all methods, as extreme pruning would render the models incapable of maintaining sufficient capacity. Thus, we limited our exploration to 90% sparsity in this study.

Table 16: Performance comparison at 90% sparsity across different methods.

| METHOD | ARC | Hella. | MMLU | TQA | Wino. | GSM8K | AVG |
|---|---|---|---|---|---|---|---|
| Mistral-7B-v0.1 | 59.98 | 83.31 | 64.16 | 42.15 | 78.37 | 37.83 | 60.97 |
| WestSeverus-7B-DPO-v2 | 71.30 | 88.26 | 63.92 | 72.72 | 83.69 | 74.27 | 75.69 |
| WildMarcoroni-Variant1-7B | 73.63 | 88.67 | 63.96 | 70.07 | 84.34 | 74.48 | 75.86 |
| Ideal Model | 73.63 | 88.67 | 63.96 | 72.72 | 84.34 | 74.48 | 76.30 |
| Task Arithmetic (Dense) | 72.52 | 89.25 | 63.39 | 74.00 | 83.46 | 73.38 | 76.00 |
| TA-dare | 70.73 | 87.18 | 60.15 | 70.69 | 82.64 | 67.93 | 73.22 (-3.08) |
| TA-magnitude | 71.47 | 89.01 | 62.74 | 73.49 | 83.48 | 72.43 | 75.44 (-0.86) |
| Ties-dare | 70.31 | 87.12 | 60.38 | 70.40 | 83.66 | 67.93 | 73.30 (-3.00) |
| Ties-magnitude | 71.57 | 88.93 | 62.71 | 73.49 | 84.08 | 73.26 | 75.67 (-0.63) |
| **CABS (Ours)** | **71.87** | **89.01** | **62.95** | **74.04** | **84.65** | **74.06** | **76.10 (-0.20)** |

