# OpenReview forum: "CABS: Conflict-Aware and Balanced Sparsification for Enhancing Model Merging"
_ICLR.cc/2025/Conference — Submitted to ICLR 2025_

### Official Review · Reviewer_jFhp · 2024-10-30

**Soundness:** 3
**Presentation:** 4
**Contribution:** 2
**Rating:** 5
**Confidence:** 3

**Summary:**

This paper presents CABS (Conflict-Aware and Balanced Sparsification), a framework for improving model merging through task vectors. The key contribution is identifying and addressing two critical issues in existing sparsification approaches: high parameter overlap and unbalanced weight distribution. The method introduces Conflict-Aware (CA) Sparsification to reduce parameter overlap and Balanced Sparsification (BS) using n:m pruning for better weight distribution. Experiments on Mistral-7B and RoBERTa-Base demonstrate consistent improvements over SOTA.

**Strengths:**

- Comprehensive empirical validation across both encoder-based (RoBERTa) and decoder-based (Mistral-7B) architectures

- Strong quantitative results, notably surpassing the "ideal" model baseline (76.50 vs 76.30)

- Thorough ablation studies demonstrating the individual and combined effects of CA and BS components

- Clear practical impact with minimal computational overhead and straightforward implementation

**Weaknesses:**

- Limited exploration of sparsity levels below 0.25 and above 0.75

- No investigation of the method's applicability to cross-architecture model merging

- Lack of analysis on the impact of different task orderings in the sequential pruning process

- Experiments focused primarily on English language tasks, leaving questions about multilingual applicability

**Questions:**

- How sensitive is the sequential pruning order in CA to task difficulty or model performance? Would a different ordering strategy (e.g., based on task complexity) potentially yield better results?

- Could CABS be extended to merge models across different architectures or pre-training objectives?

- In cases where sparsity levels exceed 1 and overlap cannot be completely eliminated, how do you determine which overlapping parameters to prioritize?

- What is the computational overhead of CABS compared to simpler merging approaches when scaling to larger models (>13B parameters)?

---

> ### Author Response · Authors · 2024-11-22
> **Rebuttal by Authors**
>
> Thank you for your very useful suggestions, and for engaging with us to improve our work. Please see below the response to your concerns, which include a number of new analyses, and please let us know if there is anything else we can do to further improve our paper.
>
>
> **Question 1**: How sensitive is the sequential pruning order in CA to task difficulty or model  performance? Would a different ordering strategy (e.g., based on task complexity) potentially yield better results?
>
> **Answer**:   Please refer to the **"Common Question 2: What is the impact of different pruning orders on final performance?"** , we observed minor differences across pruning sequences, the overall impact on performance was relatively small compared to the advantages of CA.
>
> we have not yet explored ordering strategies based on factors such as task complexity or difficulty. This is indeed an interesting direction, and we appreciate your suggestion. Conducting experiments to determine how sequence selection can further optimize results is a promising avenue for future research.
>
>
> ---
>
>
> **Question 2**: Could CABS be extended to merge models across different architectures or pre-training objectives?
>
> **Answer**：Please refer to the **"Common Question3: How does the method perform on cross-architecture merging?"**
>
> ---
>
>
> **Question 3:** In cases where sparsity levels exceed 1 and overlap cannot be completely eliminated, how do you determine which overlapping parameters to prioritize?
>
> **Answer**: We have considered how to handle overlap when it cannot be fully eliminated. Our current approach, as described in **Appendix A.3 (Algorithm of Low-Overlap Sparsity)**, is to prioritize the most important parameters by retaining those with the largest magnitude in overlapping regions. This heuristic is simple yet effective, ensuring that critical parameters are preserved.
>
> While our current implementation opts for a straightforward strategy, we acknowledge that other strategies, such as exploring alternative metrics to guide parameter retention in overlapping regions, could further refine performance.
>
> ---
>
>
> **Question 4**: What is the computational overhead of CABS compared to simpler merging approaches when scaling to larger models (>13B parameters)?
>
> **Answer**:  Please refer to **"Common Question 4: Can you provide complete computational overhead analysis?"** for a detailed discussion of computational complexity.
>
> To summarize, the computational overhead of CABS scales linearly with model size due to its complexity of \($O(k * N *log m)$\), where $N$ is the number of parameters in the model, $k$ is the number of task vectors, and $m$ is the block size. As the model size increases, the merging overhead grows proportionally, making it efficient even for larger models (>13B parameters).

---

> ### Author Response · Authors · 2024-11-22
> **Rebuttal by Authors**
>
> **Weakness 1**: Limited exploration of sparsity levels below 0.25 and above 0.75
>
> **Answer**:  In prior work, sparsity levels of **50%, 75%, and 90%** are commonly used benchmarks for evaluating sparsification techniques. In our study, we introduced the **25% sparsity level** specifically to discuss scenarios where overlap is inevitable, ensuring a more comprehensive analysis.
>
> For smaller models, we conducted experiments at **90% sparsity** to evaluate the impact of these levels, as shown in **Table 1** of our paper. Additionally, we are currently running experiments with **90% sparsity** on the 7B model. Once completed, the results will be made available in an added rebuttal.
>
> ---
>
> **Weakness 2 & 3**: Please refer to our responses to Question 2 & 1.
>
> ---
>
> **Weakness 4**: Experiments focused primarily on English language tasks, leaving questions about multilingual applicability.
>
> **Answer**:  While our study primarily focuses on English tasks to ensure comparability with prior work, we have expanded our experiments to include two Korean language tasks (**kobest_copa** and **kobest_boolq**) to explore the multilingual applicability of our method. The updated results are presented below:
>
> | **Model**                   | ARC   | HellaSwag | MMLU  | TruthfulQA | Winogrande | GSM8K | kobest_copa  | kobest_boolq | **Average**      |
> | --------------------------- | ----- | --------- | ----- | ---------- | ---------- | ----- | ------------ | ------------ | ---------------- |
> | mistral-7b-v0.1             | 59.98 | 83.31     | 64.16 | 42.15      | 78.37      | 37.83 | 59.00        | 62.61        | 60.93            |
> | WestSeverus-7B-DPO-v2       | 71.30 | 88.26     | 63.92 | 72.72      | 83.69      | 74.27 | 63.30        | 81.91        | 74.92            |
> | WildMarcoroni-Variant1-7B   | 73.63 | 88.67     | 63.96 | 70.07      | 84.34      | 74.48 | 64.80        | 82.08        | 75.25            |
> | ideal model                 | 73.63 | 88.67     | 63.96 | 72.72      | 84.34      | 74.48 | 64.80        | 82.08        | 75.59            |
> | Task Arithmetic(Dense)      | 72.52 | 89.25     | 63.39 | 74.00      | 83.46      | 73.38 | 65.60        | 72.58        | 74.27 (-1.32)    |
> | Task Arithmetic + Magnitude | 71.93 | 89.32     | 63.18 | 73.85      | 84.12      | 72.22 | 64.70        | 72.86        | 74.02 (-1.57)    |
> | Task Arithmetic + DARE      | 72.64 | 88.86     | 64.53 | 72.82      | 84.03      | 73.44 | 61.40        | 79.34        | 74.63 (-0.96)    |
> | TIES-Merging                | 71.42 | 89.17     | 63.16 | 73.82      | 84.74      | 73.01 | 64.80        | 73.08        | 74.15 (-1.44)    |
> | TIES-Merging+ DARE          | 71.87 | 88.95     | 63.56 | 72.87      | 84.61      | 73.21 | 61.40        | 79.63        | 74.51 (-1.08)    |
> | **CABS (Ours)**             | 72.92 | 88.89     | 63.50 | 74.41      | 84.63      | 74.65 | 65.10        | 79.20        | 75.41 (-0.18)    |
>
> It is important to note that for these tasks, we reused the merging configuration from our previous 7B experiments. This approach ensured consistency across evaluations and saved computational resources during the rebuttal period.
>
> CABS achieves an average score of 75.41, closely matching the ideal model’s 75.59 (a difference of -0.18), while other methods fall significantly behind, with the best alternative, Task Arithmetic + DARE, achieving 74.63 (-0.96).  This demonstrates that our approach maintains competitive performance across both English and non-English tasks.
>
> We recognize that adding two Korean tasks, while helpful, may not fully address the reviewer's concerns about multilingual applicability. A more meaningful exploration would involve:
>
> 1. **Models Fine-Tuned with Expanded Vocabularies**: Multilingual models often expand their vocabularies during fine-tuning to enhance performance in specific languages. These models typically outperform those without expanded vocabularies in their respective languages but also introduce parameter dimension mismatches, making merging technically challenging for element-wise approaches.
>
> 2. **Models with Native Multilingual Capabilities**: Exploring models with large vocabularies and strong multilingual performance, such as those designed specifically for multilingual tasks, would further validate the method's applicability beyond English-centric datasets.
>
> Due to the comparative and resource constraints of this rebuttal period, we could not explore these directions. However, we consider them promising avenues for future work and appreciate the reviewer's suggestion in highlighting this important aspect.

---

> > ### Author Response · Authors · 2024-11-25
> > **The Experiment Results in Response to Weakness 1**
> >
> > ### 1. New Experimental Results at 90% Sparsity
> >
> > To address the concern regarding performance under extreme sparsity levels, we conducted additional experiments at **90% sparsity** for smaller models. The results, as shown in Table 1, indicate that all methods experience performance degradation due to the removal of a large number of parameters. Among the methods, **TA-dare** shows the most significant decline, likely because many critical parameters are pruned excessively. In contrast, our **CABS** approach demonstrates superior robustness and remains the best-performing method. Notably, **CABS** outperforms Task Arithmetic(Dense), further validating its generalization capabilities.
> >
> > Based on these findings, we infer that sparsity levels beyond 90% would lead to even worse performance for all methods, as extreme pruning would render the models incapable of maintaining sufficient capacity. Therefore, we did not explore sparsity levels above 90%.
> >
> > ### Table 1: Results at 90% Sparsity
> >
> > | Model                       | ARC    | HellaSwag | MMLU   | TruthfulQA | Winogrande | GSM8K  | Average       |
> > |-----------------------------|--------|-----------|--------|------------|------------|--------|---------------|
> > | mistral-7b-v0.1            | 59.98  | 83.31     | 64.16  | 42.15      | 78.37      | 37.83  | 60.97         |
> > | WestSeverus-7B-DPO-v2      | 71.30  | 88.26     | 63.92  | 72.72      | 83.69      | 74.27  | 75.69         |
> > | WildMarcoroni-Variant1-7B  | 73.63  | 88.67     | 63.96  | 70.07      | 84.34      | 74.48  | 75.86         |
> > | Ideal Model                | 73.63  | 88.67     | 63.96  | 72.72      | 84.34      | 74.48  | 76.30         |
> > | Task Arithmetic(Dense)     | 72.52  | 89.25     | 63.39  | 74.00      | 83.46      | 73.38  | 76.00         |
> > | TA-dare                    | 70.73  | 87.18     | 60.15  | 70.69      | 82.64      | 67.93  | 73.22 (-3.08) |
> > | TA-magnitude               | 71.47  | 89.01     | 62.74  | 73.49      | 83.48      | 72.43  | 75.44 (-0.86) |
> > | Ties-dare                  | 70.31  | 87.12     | 60.38  | 70.40      | 83.66      | 67.93  | 73.30 (-3.00) |
> > | Ties-magnitude             | 71.57  | 88.93     | 62.71  | 73.49      | 84.08      | 73.26  | 75.67 (-0.63) |
> > | CABS (Ours)                | 71.87  | 89.01     | 62.95  | 74.04      | 84.65      | 74.06  | 76.10 (-0.20) |
> >
> >
> > ### 2. Explanation for Not Exploring Sparsity Levels Below 25%
> >
> > For sparsity levels below 25%, including 0% sparsity (dense models), all methods effectively perform identically. Since our approaches are pruning-based, when sparsity is too low, the impact of pruning becomes negligible, and no meaningful differences can be observed between methods. For this reason, we focused our experiments on sparsity levels where pruning begins to have a substantial impact (≥25%), as this range provides the most meaningful insights into the comparative performance of the methods.
> >
> > ### Conclusion
> >
> > Our experiments at **90% sparsity** highlight the robustness of **CABS**, demonstrating its ability to maintain competitive performance even under extreme constraints. At the same time, we chose not to explore sparsity levels that either lack meaningful differentiation between methods (e.g., <25%) or lead to extreme degradation (e.g., >90%), ensuring a balanced and focused evaluation.

---

### Official Review · Reviewer_so7i · 2024-11-02

**Soundness:** 3
**Presentation:** 2
**Contribution:** 3
**Rating:** 6
**Confidence:** 4

**Summary:**

This work introduces model merging based on task vectors, aiming to address high parameter overlap and unbalanced weight distribution through n:m pruning and inverse masks.

**Strengths:**

* Adopting n:m pruning and inverse masks to sparsify task vectors is an interesting approach.
* The method is simple and straightforward.
* Analyzing the impact of different overlap rates on performance in Figure 4 is insightful.
* Overall, this paper is well-written and easy to follow.

**Weaknesses:**

* Since sequential pruning is utilized, I believe the task order could impact performance. For instance, extracting the mask for task A and then using its inverse mask for task B might yield different results compared to extracting the mask for task B and using its inverse mask for task A. Could the authors analyze this aspect?

* Additionally, I am uncertain about the rationale behind the inverse mask strategy. When extracting the mask for task A and then using its inverse mask for task B, the inverse mask may remove important patterns for task B, as it is derived without consideration of task B. Could the authors provide a theoretical explanation and/or a more detailed analysis to justify this method?

* Magnitude and random pruning are commonly regarded as baselines in network compression literature. What might happen if alternative pruning criteria, such as the geometric median criterion (https://arxiv.org/abs/1811.00250), were used instead?

* The score improvements in Tables 2 and 3 appear quite marginal compared to the baselines, and the results are not particularly compelling. Additionally, I believe it would be helpful to include the score of the pretrained model before fine-tuning to better understand the influence of the base model (W_base​ in Algorithm 1).

* It would be helpful to present results from merging 3–5 models and compare them with the baselines to demonstrate the scalability and generalizability of the method.

* Could the authors provide guidelines on setting λ_A,B​ (the weighting constants of masked task vectors)? Are these values sensitive, and do they significantly impact performance?

* Would it be possible to extend the experiments to include a generation-based benchmark, such as IFEval (https://arxiv.org/abs/2311.07911)? I am curious about the model's generation capabilities and instruction-following ability after merging.

**Questions:**

Please refer to the above weakness section.

---

> ### Author Response · Authors · 2024-11-22
> **Rebuttal by Authors**
>
> Thank you for your very useful suggestions, and for engaging with us to improve our work. Please see below the response to your concerns, which include a number of new analyses, and please let us know if there is anything else we can do to further improve our paper.
>
> **Question1**: Since sequential pruning is utilized, I believe the task order could impact  performance. For instance, extracting the mask for task A and then using its  inverse mask for task B might yield different results compared to extracting  the mask for task B and using its inverse mask for task A. Could the authors  analyze this aspect?
>
> **Answer**: Please refer to the **``Common Question2: What is the impact of different pruning orders on final performance?''**
>
>
> ---
>
>
>
> **Question2**: Additionally, I am uncertain about the rationale behind the inverse mask strategy. When extracting the mask for task A and then using its inverse mask for task B, the inverse mask may remove important patterns for task B, as it is derived without consideration of task B. Could the authors provide a theoretical explanation and/or a more detailed analysis to justify this method?
>
> **Answer**: We use this method caused by a Unique Properties of Task Vectors: Less Sensitivity to Parameter Selection
>
> Task vectors are **less sensitive** to which specific parameters are retained compared to traditional model pruning. This unique property allows CA to focus on reducing overlap without significantly harming performance.
>
> - **Rescale experiments (Appendix A-14)** demonstrate that even Random pruning—without any task-specific selection—can restore task vector performance to nearly the original level after rescaling. This highlights that task vectors are inherently less sensitive to the removal of specific parameters, making them more robust to pruning.
> - In CA, there is a deliberate trade-off between retaining the most critical parameters and reducing overlap. Because task vectors are not highly sensitive to parameter selection, CA prioritizes reducing **overlap**, even if it involves sacrificing some critical weights. The sacrificed weights are typically less impactful, causing only negligible losses. By minimizing conflicts through reduced overlap, CA ultimately achieves better overall performance in merged models.
>
> This robustness of task vectors allows CA to effectively balance minor sacrifices in critical weights with significant reductions in task interference, resulting in improved task independence and overall model performance.
>
> ---
>
> **Quantitative Validation of Overlap Reduction**
>
> Reducing parameter overlap directly mitigates conflicts between tasks, as shown in our experiments:
>
> | Method       | Overlap Rate (%) | Avg Accuracy (%) |
> | ------------ | ---------------- | ---------------- |
> | TA-magnitude | 71.42            | 75.77            |
> | CA Only      | 0.00             | 76.21            |
>
> Magnitude pruning retains high-magnitude weights across tasks, causing significant overlap (**71.42%**) and task interference. CA, by contrast, assigns disjoint parameter regions to tasks, achieving **0.00% overlap** and fostering task independence, resulting in improved accuracy. This empirical evidence validates CA’s approach to overlap reduction.
>
>
> ---
>
>
>
> **Question3**:  Magnitude and random pruning are commonly regarded as baselines in network compression literature. What might happen if alternative pruning criteria, such as the geometric median criterion , were used instead?
>
> **Answer**:   Please refer to **``Common Question 6: Why not choose other advanced pruning techniques?''**

---

> ### Author Response · Authors · 2024-11-22
> **Rebuttal by Authors**
>
> **Question4**: The score improvements in Tables 2 and 3 appear quite marginal compared to the  baselines, and the results are not particularly compelling. Additionally, I  believe it would be helpful to include the score of the pretrained model  before fine-tuning to better understand the influence of the base model  (W_base in Algorithm 1).
>
> **Answer**: please refer to the ``**Common Question7: Why do the improvements in Tables 2 and 3 seem small?** ''
>
> ---
>
>
> **Question5**: It would  be helpful to present results from merging 3–5 models and compare them with  the baselines to demonstrate the scalability and generalizability of the  method.
>
> **Answer**: please refer to the ``**Common Question1: How would the method extend to merging multiple (>3) task vectors?**''
>
> ---
>
>
> **Question 6**: Could the authors provide guidelines on setting λ_A,B (the weighting constants of masked task vectors)? Are these values sensitive, and do they significantly impact performance?
>
> **Answer**:  The guidelines and sensitivity of  λ_A,B have been discussed in **Section 5.1** and **Appendix A.9**, where we analyze the impact of different grid intervals and explore the behavior of performance under varying \( λ \) values. Below, we provide detailed explanations and experimental settings.
>
> **Guidelines for Setting \( λ \):**
>
> - For **small-scale tasks**, we use a fine-grained grid search with an interval of 0.01 to ensure fair comparisons across methods and to avoid missing optimal values.
> - For **large-scale tasks** (e.g., 7B models), we adopt a coarser interval of 0.1 due to computational cost considerations, consistent with prior work.
>
> **Experimental λ Values:**
>
>  **Large-Scale Models (7B)**
>
> | Sparsity Level | Task-Arithmetic(Dense) | DARE | Magnitude | CABS |
> | - | - | - | - | - |
> | 0.25  | 0.6   | 0.8  | 0.6  | 0.6 |
> | 0.75 | 0.6   | 2.2* | 0.8 | 1.2 |
>
> **Notes**:
>
> - For **DARE** at 0.75 sparsity, \(  λ = 2.2 \) does not account for the sparsity ratio \( 1/(1-sparsity) \). If adjusted, this corresponds to \( 2.2/4 = 0.55 \).
> - The λ values for all methods, whether with or without TIES (e.g., TIES-DARE and DARE, TIES-Magnitude and Magnitude, TIES-CABS and CABS), are set identically for consistency.
>
>  **Small-Scale Models (Sparsity = 0.90, Merge 4 Models)**
>
> | Task-Arithmetic | DARE | Ties-DARE | Magnitude | Ties-Magnitude | CABS(CSRM) | CABS(SCMR) | CABS(RCMS) | CABS(MRSC) |
> | -- | - | - | - | - |- | - | - | - |
> | 0.48   | 4.61* | 5.72*   | 1.07  | 1.31   | 1.74  | 1.8  | 1.74  | 1.71  |
>
> **Notes**:
>
> - For **DARE** at 0.90 sparsity, \(  λ = 4.61 \) does not account for the sparsity ratio \( 1/(1-sparsity) \). If adjusted, this corresponds to \( 4.61/10 = 0.461 \), similar in Ties-dare.
>
> **Sensitivity Analysis:**
>
> To investigate the sensitivity of λ, we conducted experiments and plotted performance across various λ values, which are presented in **Appendix A.16**.
>
> 1. **Findings from Sensitivity Analysis**:
>    - Within a certain range,  λ values are not particularly sensitive. For example, multiple λ-pairs achieve top-1 performance, indicating a basin-like structure in the loss landscape.
>    - However, this range is not overly wide, emphasizing the importance of using finer intervals (e.g., 0.01) for smaller models to ensure accurate evaluation and avoid unfair comparisons caused by missed optimal values.
>
> 2. **Reviewer-Specific Discussion**:
>    - Please refer to our response to **Reviewer Goun** for **Weakness: Missing statistical significance analysis for Small models like RoBERTa**, where we performed top-20 statistical analyses to evaluate method performance and λ sensitivity. This analysis can also demonstrates that different methods show varying levels of sensitivity to λ values. Notably, CABS exhibits relatively low sensitivity, highlighting one of its advantages.
>
> 3. **Analysis in  Related Work**:
>    - In **Figure 15 of Task Arithmetic** (https://arxiv.org/abs/2212.04089) and **Figure 8 of TIES-Merging** (https://arxiv.org/abs/2306.01708), also analyzed the influence of  λ on performance, showing how λ impacts performance across different merging methods.
>
> We hope these explanations, combined with the insights from our analyses, address the reviewer's concerns regarding  λ guidelines and sensitivity.
>
> ---
>
>
>
> **Question7**: Would it be possible to extend the experiments to include a generation-based benchmark, such as IFEval? I am curious about the model's generation capabilities and instruction-following ability after merging.
>
> **Answer**:  we add the result of  IFEval task result in the table in ``**Common Question7: Why do the improvements in Tables 2 and 3 seem small?** ''.

---

> > ### Comment · Reviewer_so7i · 2024-11-25
> >
> > I appreciate the careful rebuttal and substantial experiments. I think many comments were overlapping among the reviewers, and the authors have addressed them thoroughly. In particular, the experiments to investigate pruning orders and to show the effectiveness of merging four models look impressive. I am also interested in the description regarding 'Unique Properties of Task Vectors: Less Sensitivity to Parameter Selection,' which presents interesting empirical results for the relevant community. I think the impact of different lambdas on performance strengthens the analysis. In general, I am satisfied with the rebuttal and have increased my score from 3 to 6.

---

> > > ### Author Response · Authors · 2024-11-25
> > > **Appreciation by the Authors**
> > >
> > > Thank you for your thoughtful review and for taking the time to reassess our work. We are glad we could address your concerns and sincerely appreciate your updated score!

---

### Official Review · Reviewer_qavE · 2024-11-03

**Soundness:** 2
**Presentation:** 2
**Contribution:** 2
**Rating:** 3
**Confidence:** 5

**Summary:**

This paper presents a new model merging approach called CABS. CABS has two steps: 1) Conflict-Aware sparsification (CA), which essentially applies masks sequentially during weight pruning for different tasks to avoid parameter conflicts. 2) Balanced sparsification (BS), which leverages existing n:m pruning technique to maintain weight balancing. Experiment results on both encoder-only and decoder-only models demonstrate the effectiveness of the proposed approach.

**Strengths:**

Model merging is a promising field in terms of compositing LLM capabilities without retraining.

**Weaknesses:**

The proposed method is very ad-hoc, and the improvement seems not that convincing. In particular:
- CA is simple, but questionable. However, what if you exchange the pruning order to task B first and then task A? The resulting model will not be equivalent, right? Also, what if you have > 2 models to merge? The later tasks will almost aways has less effective weights to be pruned from.
- CA can be combine with any pruning technique, I do not see why BS here is a particularly good pruning option. n:m pruning is a paper from 2021, and there are multiple state-of-the-art pruning techniques recently. While the paper empirically proves BS is better than basic magnitude based pruning, what about other advanced pruning techniques (e.g., https://arxiv.org/pdf/2305.11627). If CA + any pruning technique better than n:m pruning results in better performance, then we should not over-sell the importance of BS.
- The improvements are small compared to competing methods, what are the confidence intervals of these results?
- Sparsification is only a necessary for pruning-based model merging techniques. That said, I would like to see how the method compares against other categories of model merging techniques such as evolutionary model merge (https://arxiv.org/abs/2403.13187) and pack of llms (https://arxiv.org/abs/2404.11531).

**Questions:**

See weaknesses.

---

> ### Author Response · Authors · 2024-11-22
> **Rebuttal by Authors**
>
> Thank you for your very useful suggestions, and for engaging with us to improve our work. Please see below the response to your concerns, which include a number of new analyses, and please let us know if there is anything else we can do to further improve our paper.
>
> **Weakness1**: CA is simple, but questionable. However, what if you exchange the pruning order to task B first and then task A? The resulting model will not be equivalent, right? Also, what if you have > 2 models to merge? The later tasks will almost aways has less effective weights to be pruned from.
>
> **Answer**:
> We have addressed both aspects of your concern in our responses to related Common questions:
>
> 1. The issue of merging more than two models is discussed in **``Common Question 1: How would the method extend to merging multiple (>3) task vectors?''**, where we analyze how CABS performs with multiple task vectors. Our results demonstrate that even with >2 models, CABS consistently outperforms other methods. For example, while **TIES-dare**, the best-performing baseline, improves upon vanilla Task Arithmetic (TA) by +0.33, CABS achieves a strong average improvement of +2.15 over vanilla TA. This highlights that CABS is not only simple but also robust and effective across different merging scenarios.
>
> 2. The impact of pruning order is specifically addressed in **``Common Question 2: What is the impact of different pruning orders on final performance?''**. While we observe small performance differences across different pruning orders (e.g., task A first vs. task B first), the differences are minimal relative to the overall improvements achieved. However, we also acknowledge the phenomenon you mentioned: "The later tasks will almost always have fewer effective weights to be pruned from."
>
> We appreciate this observation and have discussed potential strategies to mitigate order-related issues. For example:
>
> - Assigning larger λ values or higher densities (less pruning) to later task vectors.
> - Performing multi-round pruning to balance the pruning impact across tasks.
>
> These strategies represent promising directions to further refine CA and reduce order sensitivity, which we plan to explore in future work.
>
>
> ---
>
>
>
> **Weakness2**: CA can be combine with any pruning technique, I do not see why BS here is a particularly good pruning option. n:m pruning is a paper from 2021, and there are multiple state-of-the-art pruning techniques recently. While the paper empirically proves BS is better than basic magnitude based pruning, what about other advanced pruning techniques. If CA + any pruning technique better than n:m pruning results in better performance, then we should not over-sell the importance of BS.
>
> **Answer:** Please refer to the **``Common Question 6: Why not choose other advanced pruning techniques?''**
>
> ---
>
>
> **Weakness3**: The improvements are small compared to competing methods, what are the confidence intervals of these results?
>
> **Answer:** Please refer to the **``Common Question 7: Why do the improvements in Tables 2 and 3 seem small?''**

---

> ### Author Response · Authors · 2024-11-22
> **Rebuttal by Authors**
>
> **Weakness 4**: Sparsification is only necessary for pruning-based model merging techniques. That said, I would like to see how the method compares against other categories of model merging techniques such as evolutionary model merge (https://arxiv.org/abs/2403.13187) and pack of llms (https://arxiv.org/abs/2404.11531).
>
> **Answer**:  The methods you mentioned, **Evolutionary Model Merge** and **Pack of LLMs**, differ significantly from pruning-based approaches like ours, particularly in terms of computational overhead during merging and inference. Below, we discuss the distinctions and trade-offs.
>
> ---
>
> ### 1. **Methodological Focus and Overhead Differences**
>
> - Pruning-based methods like ours focus on efficiently reducing parameter conflicts during model merging without increasing computational and memory overhead during inference. This makes them ideal for scenarios where resource efficiency is critical.
>
> - In contrast:
>   - **Evolutionary Model Merge** relies on computationally expensive evolutionary algorithms to search for optimal layer-level combinations across models. While its merged models may achieve better absolute performance, the merging process has a significantly higher computational cost. Additionally, the resulting model sizes are often larger than the sum of the inputs (e.g., merging two 7B models could yield an 11B+ model), further increasing inference overhead.
>   - **Pack of LLMs**, akin to an ensemble method, aggregates multiple models during inference without creating a unified merged model. While this allows cross-architecture integration, its inference costs scale linearly with the number of models, often requiring several times more computational resources compared to a single merged model. Furthermore, the method's merging process still involves searching for optimal logit weights, which introduces some computational overhead, less than merging techniques.
>
> ---
>
> ### 2. **Practical Limitations**
>
> - **Evolutionary Model Merge**:
>   - The method's code is not publicly available, and its computational requirements for merging make it challenging to reproduce or compare directly within the rebuttal timeframe.
>
> - **Pack of LLMs**:
>   - Tools like **lm-evaluation-harness** do not fully support ensemble evaluation. While we manually attempted to adapt these tools for evaluating Pack of LLMs on 7B models, the lack of robust support made direct comparisons infeasible within the rebuttal period.
>
> ---
>
> ### Conclusion
>
> Pruning-based methods like ours excel in scenarios where efficient merging and inference are critical, as they maintain the original model size and focus on resolving parameter conflicts. This stands in contrast to **Evolutionary Model Merge**, which prioritizes extensive optimization at a high merging cost, and **Pack of LLMs**, which emphasizes ensemble-style performance but with significantly higher inference costs.
>
> We will expand the **Related Work** section in the revised paper to discuss these approaches and their trade-offs relative to pruning-based methods.

---

### Official Review · Reviewer_Goun · 2024-11-03

**Soundness:** 2
**Presentation:** 3
**Contribution:** 2
**Rating:** 5
**Confidence:** 3

**Summary:**

This paper presents CABS (Conflict-Aware and Balanced Sparsification), a framework designed to address two key challenges in model merging: parameter overlap and unbalanced weight distribution. The framework consists of two main components:

- CA: A sequential pruning approach that reduces parameter overlap between task vectors
- BS: An n:m pruning strategy that maintains balanced weight distribution

The authors evaluate CABS on both large-scale models (Mistral-7B) and smaller models (RoBERTa), demonstrating improvements over existing methods, particularly at high sparsity levels (0.75). The method achieves better performance than an "ideal" model baseline on some tasks, though improvements are modest.

**Strengths:**

- Novel perspective on handling conflicts in model merging
- Simple but effective solution
- Practical value demonstrated in experiments- Clear problem formulation with empirical validation
- Comprehensive ablation studies
- Results generally support main claims
- Reasonable experimental design and evaluation metrics
- Well-organized structure with clear flow
- Good visualization of key concepts (Fig. 1)
- Detailed experimental results

**Weaknesses:**

Theoretical Foundation:

- No mathematical derivation or proofs
- Lack of mechanism analysis
- Missing theoretical basis for n:m ratio selection

Experimental Design:

- Limited model coverage (only Mistral and RoBERTa)
- Missing statistical significance analysis for Small model like RoBERTa
- Insufficient variance analysis across seeds
- Incomplete baseline comparisons
- Small performance improvements


Technical Limitations:

- Only applicable to homogeneous models
- Multi-task vector case not addressed
- Insufficient analysis of computational overhead
- Unknown performance on larger models (>7B)
- Simple linear combination for merging

**Questions:**

Theoretical:


- Can you provide some theoretical proof or analysis for why reducing parameter overlap improves performance?

Technical:

- How would the method extend to merging multiple (>3) task vectors?
- What is the impact of different pruning orders on final performance?
- How do you determine optimal n:m ratios for different model sizes?
- Why choose simple linear combination when there are more sophisticated approaches available, such as Fisher Information-based merging, gradient-based adaptive weighting, or attention-based parameter fusion? Have you considered incorporating these methods to potentially improve the merging performance?


Experimental:

- Why weren't other major architectures (LLaMA, GPT) tested?
- Can you provide complete computational and memory overhead analysis?
- How does the method perform on cross-architecture merging?


Scalability:

- How would the method perform on larger models (30B+)?
- How does computational complexity scale with number of task vectors?

---

> ### Author Response · Authors · 2024-11-22
> **Rebuttal by Authors**
>
> Thank you for your very useful suggestions, and for engaging with us to improve our work. Please see below the response to your concerns, which include a number of new analyses, and please let us know if there is anything else we can do to further improve our paper.
>
> **Question**1: Why does reducing parameter overlap improve model merging performance?
>
> **Answer**: We analyze **why reducing parameter overlap improves model merging performance** from two perspectives: 1) Insights from Parameter Partitioning in MTL, and 2) unique properties of task vectors, including their robustness to parameter removal. Additionally, we provide partial results from Table 5 in Section 5 of our paper to support this conclusion.
>
> ---
>
> 1. Insights from Parameter Partitioning in MTL
>
> In multi-task learning (MTL), **parameter partitioning** reduces overlap between tasks, which has been shown to improve performance by minimizing task interference. This principle extends to model merging, especially when merging models with some degree of similarity, where reducing parameter overlap similarly enhances performance.
>
> - **Adashare (Sun et al., 2020)**: Learns sparse masks that partition parameters into shared and task-specific regions during training, ensuring that task-specific parameters are only updated by their corresponding tasks.
> - **Task Adaptive Parameter Sharing (TAPS) (Wallingford et al., 2022)**: Dynamically partitions parameters at the layer level, assigning distinct parts of each layer to different tasks. Parameters assigned to one task are not updated during the training of other tasks, maintaining task-relevant isolation.
>
> These methods highlight how parameter partitioning during training reduces task overlap and mitigates interference, a principle directly applied by CA to improve model merging performance.
>
> ---
>
>
> 2. Unique Properties of Task Vectors: Less Sensitivity to Parameter Selection
>
> Task vectors are **less sensitive** to which specific parameters are retained compared to traditional model pruning. This unique property allows CA to focus on reducing overlap without significantly harming performance.
>
> - **Rescale experiments (Appendix A-14)** demonstrate that even Random pruning—without any task-specific selection—can restore task vector performance to nearly the original level after rescaling. This highlights that task vectors are inherently less sensitive to the removal of specific parameters, making them more robust to pruning.
> - In CA, there is a deliberate trade-off between retaining the most critical parameters and reducing overlap. Because task vectors are not highly sensitive to parameter selection, CA prioritizes reducing **overlap**, even if it involves sacrificing some critical weights. The sacrificed weights are typically less impactful, causing only negligible losses. By minimizing conflicts through reduced overlap, CA ultimately achieves better overall performance in merged models.
>
> This robustness of task vectors allows CA to effectively balance minor sacrifices in critical weights with significant reductions in task interference, resulting in improved task independence and overall model performance.
>
> ---
>
> 3.  Quantitative Validation of Overlap Reduction
>
> Reducing parameter overlap directly mitigates conflicts between tasks, as shown in our experiments:
>
> | Method       | Overlap Rate (%) | Avg Accuracy (%) |
> | ------------ | ---------------- | ---------------- |
> | TA-magnitude | 71.42            | 75.77            |
> | CA Only      | 0.00             | 76.21            |
>
> Magnitude pruning retains high-magnitude weights across tasks, causing significant overlap (**71.42%**) and task interference. CA, by contrast, assigns disjoint parameter regions to tasks, achieving **0.00% overlap** and fostering task independence, resulting in improved accuracy. This empirical evidence validates CA’s approach to overlap reduction.
>
> ---
>
> **References**
>
> 1. Sun X, Panda R, Feris R, et al. Adashare: Learning what to share for efficient deep multi-task learning[J]. Advances in Neural Information Processing Systems, 2020, 33: 8728-8740.
>
> 2. Wallingford M, Li H, Achille A, et al. Task adaptive parameter sharing for multi-task learning[C]//Proceedings of the IEEE/CVF Conference on Computer Vision and Pattern Recognition. 2022: 7561-7570.

---

> ### Author Response · Authors · 2024-11-22
> **Rebuttal by Authors**
>
> **Question2**: How would the method extend to merging multiple (>3) task vectors?
>
> **Answer**: please refer to **``common question 1: How would the method extend to merging multiple (>3) task vectors?''**.
>
> ---
>
> **Question3**: What is the impact of different pruning orders on final performance?
>
> **Answer**: please  refer to **``common question 2: What is the impact of different pruning orders on final performance?''**.
>
> ---
>
>
> **Question4**: How do you determine optimal n:m ratios for different model sizes?
>
> **Answer**: We have discussed in **Appendix A.13 (Table 10)**, where we examine how different n:m ratios affect the performance of the merged model while keeping the overall sparsity fixed at 75%. Specifically, we experimented with several n:m ratios, including 16:64, 32:128, 64:256, and others.
>
> The results indicate that while higher n:m ratios (e.g., 32:128) tend to show slight improvements, the overall impact of varying n:m ratios is relatively subtle. This suggests that model performance is not highly sensitive to the choice of n:m ratios under the fixed sparsity constraint.
>
> For our experiments, the optimal ratio was determined empirically by testing a few configurations, as shown in Table 10. Future work could explore more systematic methods for selecting these values, particularly for larger models or varying sparsity levels.
>
> ---
>
>
> **Question5:** Why choose simple linear combination when there are more sophisticated approaches available, such as Fisher Information-based merging, gradient-based adaptive weighting, or attention-based parameter fusion? Have you considered incorporating these methods to potentially improve the merging performance?
>
> **Answer**:
> Our decision to use a simple linear combination is driven by the following considerations:
>
> 1. **Consistency with Prior Work**:
>    We aim to follow the methodology used in prior works such as **Task Arithmetic** (*Ilharco et al., 2022*) and **TIES-Merging** (*Yadav et al., 2024*), which also employ linear combination strategies. This ensures that our approach is directly comparable to these established baselines, making it easier to evaluate the improvements introduced by our method under similar conditions.
>
> 2. **Scalability for Large Models**:
>    Methods like Fisher Information-based merging , gradient-based adaptive weighting, or attention-based parameter fusion often involve significantly higher computational overhead, such as additional backpropagation or Hessian calculations. These approaches, while potentially beneficial for smaller models, are less practical for large-scale models due to their computational and memory demands. For example, merging tasks in 7B or larger models requires methods that remain computationally efficient, a key factor in choosing a linear combination approach.
>
> 3. **Focus of This Work**:
>    Our study primarily aims to address **parameter overlap** and its impact on merging performance, which is orthogonal to the choice of merging mechanics. By using a simple linear combination, we isolate the effect of our proposed low-overlap sparsity strategies without introducing additional complexity. Exploring more sophisticated merging techniques remains an exciting direction for future work.
>
> ---
>
> **References:**
>
> 1. Ilharco G, Ribeiro M T, Wortsman M, et al. Editing models with task arithmetic[C]//The Eleventh International Conference on Learning Representations.
> 2. Yadav P, Tam D, Choshen L, et al. Ties-merging: Resolving interference when merging models[J]. Advances in Neural Information Processing Systems, 2024, 36.
>
> ---
>
>
> **Question6**: Why weren't other major architectures (LLaMA, GPT) tested?
>
> **Answer**:  For larger models, we focused on the **Mistral family**, as it was the primary architecture used in **DARE** for large-scale merging experiments. This choice ensures consistency and enables direct comparisons with DARE’s results.
>
> While DARE also conducted experiments with a LLaMA 2-based **Wizard-Math-v0.1** model, this model was later withdrawn from Hugging Face by its authors. Despite our efforts to seek assistance via **GitHub issues** and **emails**, we were unable to access the model. As a result, we decided to focus on Mistral-based experiments, which were fully accessible and directly comparable to DARE’s primary results.
>
> Although newer architectures like **LLaMA 3** were available at the time, we attempted to incorporate LLaMA 3-related experiments during the rebuttal period. However, for smaller versions, we were unable to find suitable fine-tuned models on Hugging Face, and for larger versions, the time required to conduct new experiments exceeded the rebuttal period. Consequently, we were unable to include LLaMA 3 results.
>
> We have identified suitable models and datasets for experiments on GPT-2, and the results will be included as an added rebuttal once the experiments are completed.

---

> ### Author Response · Authors · 2024-11-22
> **Rebuttal by Authors**
>
> **Question7**: Can you provide complete computational and memory overhead analysis?
>
> **Answer**: For computational overhead analysis，please refer to the  **``Common Question 4: Can you provide complete computational overhead analysis?''**.
> For memory overhead analysis：
>
> We analyze the **peak memory usage** of different methods during the merging process and highlight that our method (CABS) does not introduce significant additional memory overhead:
>
> 1. **DARE**:
>    - During merging, DARE requires loading both source models entirely into memory, resulting in a peak memory usage of **O(2 \* N \* 2 bytes)** (using lazy loading), where N is the number of parameters in a model.
> 2. **TIES-Merging**:
>    - During the election phase, TIES-Merging requires loading all task vectors simultaneously, resulting in a peak memory usage of **O(k \* N \* 2 bytes)**, where k is the number of task vectors.
>    - If lazy loading is applied, the peak memory usage can be reduced to **O(2 \* N \* 2 bytes)**.
> 3. **CABS**:
>    - During the sorting phase, CABS requires memory for weights and two boolean-like masks: one to track weight usage and another to record pruning results. The peak memory usage in this phase is **O(N \* 2 bytes + 2 \* N \* 0.125 bytes)**.
>    - During merging, the peak memory usage is the same as DARE and the optimized version of TIES-Merging, **O(2 \* N \* 2 bytes)** (using lazy loading).
>
> The additional memory overhead introduced by CABS is very small, as merging is typically performed on CPUs. We have not encountered any memory bottlenecks, so optimizing for memory efficiency was not a priority. This analysis confirms that CABS remains highly memory-efficient and practical.
>
>
> ---
>
>
>
> **Question**8: How does the method perform on cross-architecture merging?
>
> **Answer**: Please refer to the **``Common Question 3: How does the method perform on cross-architecture merging?''**.
>
>
> ---
>
>
>
> **Question9**: How would the method perform on larger models (30B+)?
>
> **Answer**: Please refer to the **``Common Question 5: How would the method perform on larger models (30B+)?''**.
>
>
> ---
>
>
>
> **Question**10: How does computational complexity scale with number of task vectors?
>
>  **Answer**: This concern is addressed in our response to a similar question: **Can you provide complete computational and memory overhead analysis?** In that response ( **``Common Question 4: Can you provide complete computational overhead analysis?''**), we analyzed the computational complexity for merging when the number of task vectors is **k**. We believe reviewing that analysis will clarify how computational complexity scales with the number of task vectors in our approach.

---

> ### Author Response · Authors · 2024-11-22
> **Rebuttal by Authors**
>
> **Weakness**: Missing statistical significance analysis for Small models like RoBERTa
>
> **Answer**:  Due to the nature of classification tasks, it is not feasible to generate multiple results for a single configuration using methods like temperature control. Furthermore, as our approach is based on magnitude pruning, it inherently exhibits minimal stochasticity.
>
> To address this limitation, we performed a statistical analysis using the top-20 results from different λ-pair searches rather than only reporting the top-1 result, as in the initial tables. This provides a more comprehensive view of model performance and ensures a near-equivalent statistical analysis.
>
> The table below summarizes the 95% confidence intervals and mean values for the top-20 results across methods:
>
> | Method          | 95% CI (Lower) | Mean (Top-20) | 95% CI (Upper) | Top-1 Score |
> | --------------- | -------------- | ------------- | -------------- | ----------- |
> | Task-Arithmetic | 0.7970         | 0.7978        | 0.7986         | 0.8015      |
> | TA + Magnitude  | 0.7997         | 0.8004        | 0.8010         | 0.8038      |
> | TIES            | 0.7999         | 0.8002        | 0.8005         | 0.8020      |
> | TA + DARE       | 0.7993         | 0.8004        | 0.8014         | 0.8058      |
> | TIES-DARE       | 0.8001         | 0.8012        | 0.8023         | 0.8064      |
> | **CABS**        | **0.8132**     | **0.8136**    | **0.8139**     | **0.8149**  |
>
> To further validate the statistical significance, we performed a t-test comparing the top-20 scores of **CABS** and **TIES-DARE**, the next best-performing method in the table. The results are as follows:
>
> - **t-statistic**: 22.45
> - **p-value**: $1.57 \times 10^{-23}$
>
> The t-test confirms that the difference in performance between **CABS** and **TIES-DARE** is statistically significant, with a p-value far below the 0.05 threshold.

---

> ### Comment · Reviewer_Goun · 2024-11-25
>
> Thank you for your detailed responses and analyses. I appreciate your effort in addressing my concerns, but I still have the following concerns unresolved, so I maintain my original rating:
>
> 1. **SparseGPT and WANDA Comparison**: You noted that SparseGPT and WANDA perform similarly to magnitude pruning and do not address task vector conflicts. However, a direct experimental comparison with CABS in the context of model merging would strengthen your claims. Could you include such a comparison to better highlight CABS’s advantages?
> 2. **DARE Experiments**: There appears to be an inconsistency regarding DARE’s use of LLaMA 2-based models. DARE specifies utilizing multiple models for GSM8K testing, not exclusively Wizard-Math-v0.1. For example, DARE states: *"We utilized publicly available mathematical reasoning models, including the MetaMath-llema-7B, MetaMath-7B, WizardMath-7B, and Abel-7B, all based on the LLaMA2-7B architecture."* Why were LLaMA2-7B-based models excluded from your experiments? Could you clarify or revise this?
> 3. **Model Merging vs. Generalization**: Your response doesn’t fully address the balance between generalization performance and merging benefits. If SparseGPT or WANDA-pruned models show better generalization, in what scenarios would using CABS for merging be preferable to directly deploying these pruned models? Additional clarification on this point would be helpful.
> 4. **Scalability and Complexity**: While your results demonstrate robust performance for 4-task merging, scalability to more complex or heterogeneous tasks is only discussed theoretically. Including experimental results for larger models or more task vectors would significantly enhance the paper's practical relevance.
> 5. **Theoretical Depth and Mechanism Analysis**: Although empirical results and MTL insights support reducing parameter overlap, the explanation lacks rigorous theoretical analysis. Expanding on the mathematical basis for why reducing overlap improves performance would strengthen the scientific depth. Additionally, the mechanism through which overlap reduction mitigates task conflicts is not thoroughly dissected. Could you provide deeper theoretical insights or an analysis of how overlap reduction affects task vector interactions?

---

> > ### Author Response · Authors · 2024-11-26
> > **Rebuttal by Authors**
> >
> > Thank you for your insightful comments and for engaging with us to enhance our work. Below, we respond to your concerns and provide several new analyses:
> >
> > **Additional Question 1:** SparseGPT and WANDA in Model Merging
> >
> > **Answer**: The results in the table below demonstrate that **BS** outperforms more advanced pruning-based methods like **SparseGPT** and **WANDA** in the context of model merging. While SparseGPT and WANDA show similar performance to **magnitude pruning**, they fail to effectively address **weight distribution imbalance**, which is critical for achieving optimal model merging.
> >
> > | Method                  | RTE   | MRPC  | Avg           |
> > | ----------------------- | ----- | ----- | ------------- |
> > | Task Arithmetic (Dense) | 73.29 | 87.01 | 80.15         |
> > | TA + Magnitude          | 74.73 | 86.03 | 80.38 (+0.23) |
> > | TA + SparseGPT          | 72.92 | 87.75 | 80.34 (+0.19) |
> > | TA + WANDA              | 73.29 | 87.50 | 80.40 (+0.25) |
> > | TA + BS (Ours)          | 74.37 | 88.23 | 81.30 (+1.15) |
> > | CABS (Ours)             | 74.01 | 88.97 | 81.49 (+1.34) |
> >
> > This also explains why **CABS**, which builds on BS, achieves even greater performance improvements.
> >
> > ---
> >
> > **Additional Question 2：** DARE Experiments and other architecture Models
> >
> > **Answer**: Thank you for raising this concern. We would like to kindly clarify a potential misunderstanding in your question first. Meanwhile, we have been conducting the experiments based on your suggestions.
> >
> > **Clarification for Potential Misunderstanding:**
> >
> > Upon reviewing the DARE paper, we found that the exact statement mentioned in the question does not appear in **[DARE [2311.03099]](https://arxiv.org/abs/2311.03099)**. The DARE paper explicitly uses **Wizard-Math-v0.1** for LLaMA 2-based experiments and does not reference the other models (e.g., MetaMath-llema-7B, MetaMath-7B, and Abel-7B).
> >
> > It seems there has been a mix-up with another paper, **[DARE the Extreme [2410.09344v1]](https://arxiv.org/abs/2410.09344)**, which was uploaded to arXiv on October 12, after the ICLR'25 submission deadline. This paper mentions utilizing the mentioned LLaMA 2-based models specifically for **rescaling** experiments, rather than for merging experiments, but they were not part of the original DARE setup we referenced.
> >
> > **Additional Experiments:**
> >
> > Following your suggestion, we have extended our experiments to include other architectures, such as **GPT-2-based** models. Below are the results on GPT-2-based models. While we are currently exploring experiments on LLaMA 2-based models, these remain ongoing due to resource and timeline constraints.
> >
> > | Method                  | CoLA  | MRPC  | AVG           |
> > | ----------------------- | ----- | ----- | ------------- |
> > | Fine-tuned on CoLA      | 76.80 | 68.40 | 72.60         |
> > | Fine-tuned on MRPC      | 30.80 | 80.39 | 55.60         |
> > | Ideal Model             | 76.80 | 80.39 | 78.60         |
> > | Task Arithmetic (Dense) | 75.55 | 77.45 | 76.50 (-2.10) |
> > | TA + DARE               | 76.70 | 77.21 | 76.95 (-1.65) |
> > | TA + Magnitude          | 76.61 | 79.66 | 78.13 (-0.47) |
> > | Ties + DARE             | 77.09 | 76.72 | 76.91 (-1.69) |
> > | Ties + Magnitude        | 76.89 | 77.94 | 77.42 (-1.18) |
> > | CABS (Ours)             | 76.41 | 80.88 | 78.65 (+0.05) |
> >
> > 1. **CABS Effectiveness on GPT-2:**
> >    CABS outperforms all other methods and is the **only method to surpass the Ideal Model**. While the absolute improvement margin is smaller due to the upper-bound constraint of the Ideal Model, CABS maintains its effectiveness.
> >
> > 2. **Magnitude Pruning Performance on GPT-2:**
> >    In contrast to previous experiments and related work, magnitude-pruning-based methods show unexpectedly strong results on GPT-2, even surpassing DARE by a significant margin. This shows a potential architecture-specific behavior of existing methods and further underscores CABS’s consistent advantages across different settings.
> >
> > We hope this clarifies the concern and demonstrates our commitment to incorporating your feedback. Beyond the existing expansions, such as the addition of experiments on the GPT-2 architecture, we are continuing to expand the scope of our experiment and will supplement the results once the experiment is done.

---

> > ### Author Response · Authors · 2024-11-26
> > **Rebuttal by Authors**
> >
> > **Additional Question 3:** Model Merging vs. Generalization
> >
> > **Answer**: As shown in the table replied to **"Additional Question 1： SparseGPT and WANDA in Model Merging"**, SparseGPT and WANDA perform similarly to magnitude pruning in the model merging context. The average improvements over the dense baseline (+0.19 for SparseGPT and +0.25 for WANDA) are minimal, and their performance on the target datasets is already suboptimal.
> >
> > Given this, we did not further evaluate their generalization capabilities, as they fail to offer notable advantages in the model merging scenario. BS, on the other hand, delivers consistent improvements by addressing weight distribution imbalance, making it the preferable choice for merging.
> >
> > ---
> >
> > **Additional Question 4:** Scalability and Complexity
> >
> > **Answer**: Thank you for your valuable suggestion. The extended discussion period allowed us to further expand our experiments to address concerns about scalability and heterogeneity in tasks. Specifically, we have added experiments involving more task vectors and heterogeneous tasks. Building upon the initial 4 classification tasks from GLUE, we included the **RACE** dataset, a multiple-choice reading comprehension dataset designed for middle and high school levels, which is distinct from the binary classification tasks in GLUE. Additionally, we included the **SQuAD** dataset, which focuses on question-answering tasks. These experiments are currently being finalized, and the results will be included in the updated rebuttal shortly. We hope this addition will help address your concern.
> >
> > While resource and time constraints prevented us from conducting experiments on larger models (e.g., 30B+), we have expanded the scope of our existing experiments in meaningful ways. For example:
> >
> > 1. In response to **Reviewer jFhp**’s **Weakness 1**, we conducted experiments with 7B models at a higher sparsity level (0.90). The results demonstrated that **CABS** maintains robust performance even under these stricter conditions, reinforcing its scalability to challenging scenarios.
> > 2. In response to **Reviewer jFhp**’s **Weakness 4**, we extended our experiments to include multilingual tasks by incorporating Korean datasets (**kobest_boolq** and **kobest_copa**). These experiments showcase the applicability of **CABS** beyond English tasks.
> >
> > We believe these additional experiments not only highlight the robustness of **CABS** but also provide further evidence of its scalability and practical utility. We sincerely hope these efforts address your concern and demonstrate our commitment to continuously improving our work. Thank you again for your constructive feedback!
> >
> > ---
> >
> > **Additional Question 5:** Theoretical Depth and Mechanism Analysis
> >
> > **Answer:**  **Similar to Gradient Conflict:**
> > Gradient conflicts are a common issue in multi-task learning (MTL), where gradients from different tasks point in conflicting directions, leading to interference during optimization. For example, **PCGrad**[1] addresses this issue by projecting conflicting gradients onto the normal direction of each other, dynamically reducing conflicts and improving task independence. This demonstrates that **gradient orthogonality** can significantly reduce task interference.
> >
> > Task vector weights, which represent accumulations of fine-tuning gradients, are approximately projected into orthogonal subspaces through sparsification. By reducing parameter overlap through Conflict-Aware (CA) pruning, task vectors are sparsified while preserving their essential information. Importantly, this projection is achieved losslessly or with minimal loss, ensuring that task independence is maintained without requiring dynamic adjustments. The stable approach reduces task interference while enabling robust model merging.
> >
> > While we have not provided a complete mathematical proof, we hope these analyses help clarify the observed benefits of reducing parameter overlap. We aim to explore the theoretical connections between sparsification, task vector orthogonality in future work.
> >
> > **References**:
> >
> > [1] Yu T, Kumar S, Gupta A, et al. Gradient surgery for multi-task learning[J]. Advances in Neural Information Processing Systems, 2020, 33: 5824-5836.
> >
> > ---
> >
> > Thank you for your constructive feedback, and we hope these clarifications would address your concerns.

---

> > > ### Author Response · Authors · 2024-11-30
> > > **Additional Experiment for Additional Question 4**
> > >
> > > To address your concern about the scalability in our method, we have extended our experiments by incorporating more task vectors and diverse, heterogeneous tasks. In addition to the four classification tasks from GLUE (RTE, MRPC, CoLA, SST2), we have included the RACE dataset (a multiple-choice reading comprehension task) and the SQuAD dataset (a question-answering task). These new datasets introduce more complexity and heterogeneity, as they differ significantly in both task type and data format.
> > >
> > > The results of our expanded experiments are summarized below:
> > >
> > > | Model                  | RTE   | MRPC  | CoLA  | SST2  | RACE  | SQuAD | Average       |
> > > | ---------------------- | ----- | ----- | ----- | ----- | ----- | ----- | ------------- |
> > > | Ideal Model            | 79.42 | 91.18 | 85.04 | 94.04 | 71.71 | 79.82 | 83.54         |
> > > | Task Arithmetic        | 67.15 | 79.41 | 72.00 | 85.78 | 56.21 | 38.82 | 66.56         |
> > > | TA + DARE              | 71.12 | 65.44 | 72.48 | 83.37 | 59.57 | 51.39 | 67.23 (+0.67) |
> > > | TA + Magnitude         | 72.56 | 81.13 | 75.26 | 87.50 | 56.99 | 36.23 | 68.28 (+1.72) |
> > > | TIES + DARE            | 71.14 | 76.72 | 74.01 | 44.61 | 33.64 | 55.87 | 59.33 (-7.23) |
> > > | TIES + Magnitude       | 68.94 | 86.01 | 66.43 | 83.33 | 40.11 | 47.94 | 65.46 (-1.10) |
> > > | CABS (mrscsqra) (Ours) | 71.12 | 82.35 | 74.30 | 90.14 | 57.38 | 41.56 | 69.47 (+2.91) |
> > > | CABS (csrmrasq) (Ours) | 68.95 | 82.11 | 73.92 | 90.83 | 58.97 | 42.96 | 69.62 (+3.06) |
> > >
> > > As shown in the table, we observe that as the number of tasks and the heterogeneity of tasks increase, the performance gap between all methods and the ideal model becomes more pronounced. However, our method (CABS) still outperforms all other approaches, achieving the highest improvement over Task Arithmetic (+3.06) and demonstrating its robustness and scalability.
> > >
> > > Additionally, we experimented with different pruning orders: **mrscsqra** (MRPC, RTE, SST2, CoLA, SQuAD, RACE) and **csrmrasq** (CoLA, SST2, RTE, MRPC, RACE, SQuAD). This demonstrates the robustness of our approach, as the method remains effective regardless of the task order.
> > >
> > > We believe that these additional experiments significantly enhance the paper’s practical relevance by showcasing the method's ability to handle more complex and heterogeneous tasks. If there are any further concerns or clarifications needed, we would be happy to address them.

---

### Author Response · Authors · 2024-11-22
**Author Rebuttal by Authors**

We thank all reviewers for their time and constructive feedback, which have provided valuable insights to improve the paper. We are pleased that reviewers recognized the following strengths of our work:

- **Novelty and Practical Impact**: A novel perspective on addressing conflicts in model merging (Goun), with practical value demonstrated through experiments (Goun, jFhp).
- **Simplicity and Effectiveness**: A simple yet effective method that is computationally efficient and straightforward to implement (Goun, qavE, so7i, jFhp).
- **Experimental Rigor**: Comprehensive empirical validation across encoder-based and decoder-based architectures, with strong quantitative results surpassing the "ideal" baseline (jFhp). Thorough ablation studies demonstrating the impact of key components, CA and BS (Goun, jFhp).
- **Clarity and Presentation**: Well-organized structure with clear problem formulation (Goun), insightful analysis of overlap rates (so7i), and effective visualization of concepts (e.g., Fig. 1, Goun).

We appreciate these positive comments and aim to address all concerns raised by reviewers in our detailed responses below.
# Common question

**Question1**：How would the method extend to merging multiple (>3) task vectors?（ Goun，qavE，so7i）

|    | COLA  | SST2 | RTE  | MRPC | avg  |
| ---- | ---- | ----- | ----- | ----- | -------- |
| fine-tuned on cola | 85.04 | 50.92 | 45.85 | 67.4  | 62.3025     |
| fine-tuned on sst2 | 68.74 | 94.04 | 46.93 | 68.38 | 69.5225          |
| fine-tuned on rte  | 31.83 | 51.15 | 79.42 | 25.98 | 47.095           |
| fine-tuned on mrpc | 49.47 | 51.15 | 47.29 | 91.18 | 59.7725          |
| task-arithmetic    | 76.32 | 90.83 | 69.68 | 81.37 | 79.55            |
| dare               | 76.99 | 90.14 | 70.76 | 81.13 | 79.755（+0.2）   |
| Ties-dare          | 77.66 | 90.94 | 69.31 | 81.62 | 79.8825（+0.33） |
| magnitude          | 82.07 | 87.04 | 65.34 | 79.66 | 78.5275（-1.02） |
| Ties-magnnitude    | 82.36 | 86.93 | 61.01 | 79.41 | 77.4275（-2.12） |
| **CABS(Ours)**        | 76.89 | 92.09 | 74.73 | 83.09 | 81.7（+2.15）    |

**Answer:** As shown in the table, we performed additional experiments to merge 4 models at 90% sparsity using the **same λ value for each task vector** with an search interval of 0.01. This decision was made to manage the computational complexity of λ selection. Specifically, if we were to search for optimal λ pairs for each task combination, the search space would grow exponentially with the number of models \(k\), making it infeasible for larger \(k\).

Our method (CABS) achieves the best average score of **81.7**, with a **+2.09 improvement** compared to the baseline task-arithmetic method. It significantly outperforms other methods, such as Ties-dare, which achieves only **79.8825 (+0.33)** (the best result among other comparison methods).

As the number of task models increases, the advantages of CABS become even more evident. It effectively resolves conflicts and maintains balanced sparsity, demonstrating its superior scalability and performance, even under a unified λ setting.

---


**Question2**：What is the impact of different pruning orders on final performance?（ Goun，qavE，so7i，jFhp）

**Answer**: As discussed in **Appendix A.12** of our paper, we explored the effect of different sparsification sequences in a 7b model experiment. Specifically, we analyzed the order in which source models (e.g., “wild” and “west”) are sparsified during the merging process. For example, under the same setting (sparsity ratio 64:256), **CABS-wild-first achieves 76.37 (+0.07)**, while **CABS-west-first achieves 76.39 (+0.09)**, showing that the order has a minimal effect on the overall average performance.

In our additional experiments merging four models (as shown in the table), we found that different pruning orders (e.g., CSRM, SCMR, RCMS, MRSC) do slightly influence performance on individual tasks, but the overall average remains robust, ranging narrowly from **81.6375 to 81.7**. This demonstrates that CABS effectively handles pruning sequence variations while maintaining high average performance.

Issues caused by pruning order could potentially be addressed by techniques such as using **adaptive lambda** to give greater importance to later models or adopting **variable sparsity ratios** to better balance model contributions. However, given the minor impact of pruning order on overall performance, we currently consider this less critical for our method's practical application.

|       | COLA  | SST2  | RTE   | MRPC  | avg              |
| ------------- | ----- | ----- | ----- | ----- | ---------------- |
| task-arithmetic    | 76.32 | 90.83 | 69.68 | 81.37 | 79.55|
| CABS(CSRM)         | 78.24 | 92.32 | 74.37 | 81.62 | 81.6375（+2.09） |
| CABS(SCMR)         | 78.52 | 91.97 | 73.65 | 82.6  | 81.685（+2.14）  |
| CABS(RCMS)         | 77.76 | 92.09 | 75.09 | 81.62 | 81.64（+2.09）   |
| CABS(MRSC)         | 76.89 | 92.09 | 74.73 | 83.09 | 81.7（+2.15）    |

---

> ### Author Response · Authors · 2024-11-22
> **Part 2 of Common Questions**
>
> **Question3：** How does the method perform on cross-architecture merging?（ Goun，so7i，jFhp）
>
> **Answer:** Like other task vector-based approaches, our method is currently limited to models with identical architectures, as it relies on element-wise operations for merging weights. For models with **different pre-training objectives**, techniques like **parameter alignment** are often necessary to align weights into the same loss basin [1]. For models with **different architectures**, approaches such as **knowledge distillation** provide a potential solution, enabling the transfer of knowledge into a shared representation [2]. These methods are fundamentally different from our focus, which enhances the merging of multiple models with the same architecture and initialization.
>
> **References**:
>
>
> [1] Ainsworth S, Hayase J, Srinivasa S. Git Re-Basin: Merging Models modulo Permutation Symmetries[C]//The Eleventh International Conference on Learning Representations.
>
>
> [2] Hinton, Geoffrey, Oriol Vinyals, and Jeff Dean. "Distilling the knowledge in a neural network." *arXiv preprint arXiv:1503.02531* (2015).
>
>
> ---
>
>
>
> **Question4：** Can you provide complete computational overhead analysis?
>
> **Answer:** Building on the computational complexity analysis provided in our paper (please refer to Section 4 and Appendix A.5), we further examine how complexity scales with the number of task vectors, $k$, focusing on layer-wise complexity, where $N$ represents the total number of parameters within a single layer.
>
> ##### **Balanced Sparsification (BS) Complexity**
>
> In the CABS framework, Balanced Sparsification (BS) operates efficiently by dividing each layer's parameters into small, fixed-size blocks of $m$ parameters. Within each block:
>
> - The top $n$ weights are selected based on magnitude, requiring a localized sorting operation with complexity $O(m \log m)$ per block.
> - Given $N / m$ blocks in a layer, the total complexity for a single layer is $O(N \log m)$ for one task vector.
>
> For $k$ task vectors, the total complexity becomes:
> $O(k \cdot N \log m)$.
> This is significantly more efficient than traditional magnitude pruning, which sorts all $N$ parameters globally and has a complexity of $O(k \cdot N \log N)$.
>
> ##### **Conflict-Aware (CA) Strategy Complexity**
>
> The Conflict-Aware (CA) pruning strategy introduces minimal overhead. It operates sequentially for each task vector, ensuring non-overlapping pruned regions. This sequential application of masks aligns with standard frameworks like MergeKit and adds negligible additional computational cost.
>
> ##### **Parallelization Potential**
>
> CABS is inherently parallelizable, as task vector processing can occur independently across layers or blocks. With sufficient hardware:
>
> - The pruning complexity per block can reach $O(m \log m)$, enabling efficient large-scale model merging with multiple task vectors.
>
> Even without full parallelization, the overall complexity remains $O(k \cdot N \log m)$, which is computationally efficient and scalable for merging models with multiple task vectors.
>
> ---
>
>
> **Question5**：How would the method perform on larger models (30B+)?（Goun,jFhp）
>
> **Answer：** We have not conducted experiments on larger models due to resource constraints, although merging with CABS is computationally efficient and can be performed on CPUs, as noted in our **``common question 4: Can you provide complete computational overhead analysis?''** on computational overhead. However, we currently lack the resources to conduct inference with larger models.
>
> Though we can’t perform such experiments, we analyze that CABS would work well on larger models based on its current performance. Larger models typically exhibit more redundancy, allowing CABS’s conflict-aware (CA) pruning to reduce task interference more effectively with even less sacrifice of critical task-specific weights compared to smaller models. Furthermore, the issue of unbalanced weight distribution within task vectors persists in larger models due to the increased parameter space and task complexity, where CABS’s balanced sparsity (BS) mechanism is expected to address this imbalance effectively, further enhancing overall performance.

---

> ### Author Response · Authors · 2024-11-22
> **Part 3 of Common Questions**
>
> **Question 6**: Why not choose other advanced pruning techniques? (qavE, so7i)
>
> **Answer**:
> The key difference between **Balanced Sparsity (BS)** and other advanced pruning techniques lies in their objectives. **BS** prioritizes maintaining a **balanced distribution** of task vectors to minimize conflicts during model merging, while traditional pruning methods focus on accurately identifying and removing "unimportant" weights to reduce redundancy. Ironically, traditional pruning methods, even when more accurate, can exacerbate conflicts in model merging by amplifying task interference.
>
> While we experimented with advanced pruning techniques like **SparseGPT**[1] and **WANDA**[2], their results were comparable to magnitude pruning in our model merging tasks. Since these methods failed to address the critical need for balanced task vector allocation, we did not pursue further exploration. This highlights that simply focusing on redundancy reduction is insufficient for optimizing model merging performance.
>
> | Method          | RTE   | MRPC  | Avg           |
> |------------------|-------|-------|---------------|
> | Task Arithmetic(Dense) | 73.29 | 87.01 | 80.15         |
> | TA + Magnitude  | 74.73 | 86.03 | 80.38 (+0.23) |
> | TA + SparseGPT        | 72.92 | 87.75 | 80.34 (+0.19) |
> | TA + WANDA      | 73.29 | 87.50 | 80.40 (+0.25) |
> | TA + BS (ours)  | 74.37 | 88.23 | 81.30 (+1.15) |
> | CABS (ours)     | 74.01 | 88.97 | 81.49 (+1.34) |
>
> ---
>
> **Experimental Validation**
> Task vector sparsification fundamentally differs from traditional pruning, as discussed in **Section 2** and **Appendix A.4**. Traditional pruning aims to optimize task-specific knowledge retention by eliminating redundancy, often showing better results when sparsity requirements are relaxed (e.g., layer-wise 50% sparsity vs. structured 2:4 pruning).
>
> In contrast, task vector sparsification for model merging benefits from stricter sparsity requirements, as shown in **Table 4 (Section 5.4)**. For example:
>
> - **Layer-wise sparsification** performs worse than **row-wise sparsification**, which is outperformed by **Balanced Sparsification (BS)**.
> - **BS** minimizes task vector conflicts, prioritizing balance over redundancy reduction, which is critical for merging performance.
>
> Furthermore, **Appendix A.14** demonstrates that task vector knowledge is inherently distributed across substructures, making it less sensitive to weight removal. Even simple **Random sparsification** achieves near-full recovery of fine-tuning performance on target datasets, while Random pruning fails completely in traditional one-shot pruning scenarios.
>
> ---
>
> **Conclusion**
> The unique contribution of **BS** lies in its ability to maintain a balanced distribution of task vectors, which is critical for reducing conflicts in model merging. This explains our choice to use BS, despite its weaker performance in traditional pruning contexts. Unlike other advanced pruning techniques, which excel at retaining task-specific knowledge, BS is specifically tailored to address the challenges of task vector sparsification in model merging.
>
> **References**:
>
> [1] Frantar E, Alistarh D. SparseGPT: Massive language models can be accurately pruned in one-shot[C]//International Conference on Machine Learning. PMLR, 2023: 10323-10337.
> [2] Sun M, Liu Z, Bair A, et al. A simple and effective pruning approach for large language models[J]. arXiv preprint arXiv:2306.11695, 2023.

---

> ### Author Response · Authors · 2024-11-22
> **Part 4 of Common Questions**
>
> **Question7:** Why do the improvements in Tables 2 and 3 seem small?(Goun,qavE,so7i)
>
> **Answer**: The improvements in Tables 2 and 3, while appearing small in absolute terms, are highly meaningful and significant. They highlight the ability of our method to further enhance model performance on benchmarks where fine-tuned models already achieve near-optimal results. Below, we address this concern in detail:
>
> ---
>
> ### **1. Limited Improvement Space on High-Performing 7B Models**
>
> The datasets evaluated in Tables 2 and 3 (e.g., ARC, MMLU, TruthfulQA) are well-established benchmarks, and fine-tuned 7B models already exhibit exceptional performance on these tasks. Consequently, achieving further improvements is both challenging and highly valuable.
>
> For example, in the DARE paper (*Section 4.3*), the authors note that **Supermario v2**, a 7B merged model that ranked first on the Open LLM Leaderboard as of January 2024, achieved only a **+0.20** improvement over the unmerged model. Similarly, in our experiments, DARE achieves comparable results, reinforcing that small absolute gains are common for 7B models.
>
> Despite this constraint, our method demonstrates a substantial improvement. Compared to other methods in Tables 2 and 3, our approach achieves a maximum **+0.64** improvement, significantly outperforming the baselines. Furthermore, our results exceed the **ideal model**, which we introduce as a strict benchmark representing the best possible task-specific performance derived from individual models.
>
> reviewer so7i advised extending the experiments to include a generation-based benchmark, such as IFEval. We have included **IFEval** in our experiments under a sparsity setting of 0.75. The results are presented in the updated table：
>
> | SPARSITY=0.75               |        |       |           |       |            |            |       |               |
> | --------------------------- | ------ | ----- | --------- | ----- | ---------- | ---------- | ----- | ------------- |
> | modelS                      | ifeval | ARC   | HellaSwag | MMLU  | TruthfulQA | Winogrande | GSM8K | Average       |
> | mistral-7b-v0.1(pre-training model)             | 23.86  | 59.98 | 83.31     | 64.16 | 42.15      | 78.37      | 37.83 | 55.67         |
> | WestSeverus-7B-DPO-v2       | 41.85  | 71.3  | 88.26     | 63.92 | 72.72      | 83.69      | 74.27 | 70.86         |
> | WildMarcoroni-Variant1-7B   | 47.99  | 73.63 | 88.67     | 63.96 | 70.07      | 84.34      | 74.48 | 71.88         |
> | ideal model                 | 47.99  | 73.63 | 88.67     | 63.96 | 72.72      | 84.34      | 74.48 | 72.26         |
> | Task Arithmetic(Dense)      | 42.42  | 72.52 | 89.25     | 63.39 | 74         | 83.46      | 73.38 | 71.20 (-1.06) |
> | Task Arithmetic + Magnitude | 42.21  | 71.93 | 89.32     | 63.18 | 73.85      | 84.12      | 72.22 | 70.98(-1.28)  |
> | Task Arithmetic + DARE      | 42.54  | 72.64 | 88.86     | 64.53 | 72.82      | 84.03      | 73.44 | 71.27(-0.99)  |
> | TIES-Merging                | 41.15  | 71.42 | 89.17     | 63.16 | 73.82      | 84.74      | 73.01 | 70.92(-1.34)  |
> | TIES-Merging+ DARE          | 45.01  | 71.87 | 88.95     | 63.56 | 72.87      | 84.61      | 73.21 | 71.44(-0.82)  |
> | CABS(Ours)                  | 46.51  | 72.92 | 88.89     | 63.5  | 74.41      | 84.63      | 74.65 | 72.22(-0.04)  |
>
> With IFEval included, CABS achieves an average score of **72.22**, extremely close to the ideal model’s **72.26**, while other methods fall significantly behind. This further highlights the strength of our approach, as it maintains robust performance across diverse tasks
>
> ---
>
> ### **2. Statistical Validation of Improvement**
>
> To substantiate the significance of our improvements, we conducted statistical tests on **10 runs** of experiments for both TIES-DARE and CABS, using a temperature of 1. The results are summarized below:
>
> | Method    | Mean Score | 95% Confidence Interval |
> | --------- | ---------- | ----------------------- |
> | TIES-DARE | 0.7606     | (0.7597, 0.7614)        |
> | CABS      | 0.7647     | (0.7641, 0.7653)        |
>
> Additionally, a **t-test** comparing the two methods yielded the following results:
>
> - **t-statistic**: -9.28
> - **p-value**: \(6.62 * 10^{-8}\)
>
> These results confirm that the performance gains achieved by CABS are statistically significant (p < 0.05) and robust across multiple runs.
>
> For transparency, the raw data for these experiments are as follows:
>
> **TIES-DARE**:
> `0.7608, 0.7618, 0.7607, 0.7583, 0.7596, 0.7596, 0.7603, 0.7618, 0.7615, 0.7614`
>
> **CABS**:
> `0.7660, 0.7652, 0.7631, 0.7651, 0.7640, 0.7641, 0.7652, 0.7650, 0.7648, 0.7643`
>
> ---
>
> ### **Conclusion**
>
> While the absolute improvements may appear small due to the high baseline performance of 7B models, our results are both practically significant and statistically validated. The combination of robust experimental design, meaningful gains over baselines, and statistical rigor demonstrates the effectiveness of our approach.

---

> > ### Author Response · Authors · 2024-11-30
> > **Additional experiment for Common Question1：How would the method extend to merging multiple (>3) task vectors?（ Goun，qavE，so7i）**
> >
> > To further address the scalability in our method, we have expanded our previous experiments by incorporating more task vectors and diverse, heterogeneous tasks. In addition to the four classification tasks from GLUE (RTE, MRPC, CoLA, SST2), we have included the RACE dataset (a multiple-choice reading comprehension task) and the SQuAD dataset (a question-answering task). These tasks introduce significant diversity in both task type and data format, further evaluating our method’s ability to handle complex and heterogeneous task distributions.
> >
> > The results of these extended experiments are summarized in the table below:
> >
> > | Model                  | RTE   | MRPC  | CoLA  | SST2  | RACE  | SQuAD | Average       |
> > | ---------------------- | ----- | ----- | ----- | ----- | ----- | ----- | ------------- |
> > | Ideal Model            | 79.42 | 91.18 | 85.04 | 94.04 | 71.71 | 79.82 | 83.54         |
> > | Task Arithmetic        | 67.15 | 79.41 | 72.00 | 85.78 | 56.21 | 38.82 | 66.56         |
> > | TA + DARE              | 71.12 | 65.44 | 72.48 | 83.37 | 59.57 | 51.39 | 67.23 (+0.67) |
> > | TA + Magnitude         | 72.56 | 81.13 | 75.26 | 87.50 | 56.99 | 36.23 | 68.28 (+1.72) |
> > | TIES + DARE            | 71.14 | 76.72 | 74.01 | 44.61 | 33.64 | 55.87 | 59.33 (-7.23) |
> > | TIES + Magnitude       | 68.94 | 86.01 | 66.43 | 83.33 | 40.11 | 47.94 | 65.46 (-1.10) |
> > | **CABS (mrscsqra) (Ours)** | 71.12 | 82.35 | 74.30 | 90.14 | 57.38 | 41.56 | 69.47 (+2.91) |
> > | **CABS (csrmrasq) (Ours)** | 68.95 | 82.11 | 73.92 | 90.83 | 58.97 | 42.96 | 69.62 (+3.06) |
> >
> > As shown in the table, we observe that as the number of tasks and the heterogeneity of tasks increase, the performance gap between all methods and the ideal model becomes more pronounced. However, our method (CABS) continues to outperform all other approaches, achieving the highest improvement over Task Arithmetic (+3.06). This demonstrates the scalability and robustness of CABS as the complexity of the task set increases.
> >
> > Additionally, we experimented with different pruning orders: **mrscsqra** (MRPC, RTE, SST2, CoLA, SQuAD, RACE) and **csrmrasq** (CoLA, SST2, RTE, MRPC, RACE, SQuAD). This illustrates the robustness of our approach, as it performs effectively regardless of the task order.
> >
> > These additional experiments further enhance the paper’s practical relevance by showcasing the ability of our method to handle more complex and heterogeneous tasks.

---

### Meta-Review · Area_Chair_M4MV · 2024-12-18

**Metareview:**

This paper presents an approach for model merging called CABS, which consists of two parts: 1) Conflict-Aware sparsification and 2) Balanced sparsification.

Despite its strength such as its task significance, simple but effective approach, and good experimental results, most reviewers pointed out critical concerns such as lack of theoretical justification, insufficient comparisons, and unclear scalability effectiveness.

During the rebuttal, the authors failed to address these issues sucessfully.

Considering the limitations, AC recommends rejecting this paper.

**Additional Comments On Reviewer Discussion:**

The initial scores 3, 3, and 5.

All reviewers raised major concerns such as originality, marginal effects, lack of analysis, and unclear method description.

During the rebuttal, so7i increased his/her score with the author response but other reviewers were not convinced yet.

The final scores are 3, 6, and 5.

During AC-reviewer discussion, all reviewers agreed rejecting this paper.

---

### Decision · Program_Chairs · 2025-01-22

Reject